# TNF is a potential therapeutic target to suppress prostatic inflammation and hyperplasia in autoimmune disease

Renee E. Vickman [1], LaTayia Aaron-Brooks[1,2], Renyuan Zhang [3], Nadia A. Lanman [4,5], Brittany Lapin [6,9], Victoria Gil [1], Max Greenberg[1], Takeshi Sasaki[1,10], Gregory M. Cresswell[4,11], Meaghan M. Broman[4], J. Sebastian Paez [5,7], Jacqueline Petkewicz[1], Pooja Talaty[1], Brian T. Helfand[1], Alexander P. Glaser [1], Chi-Hsiung Wang[1,6], Omar E. Franco[1], Timothy L. Ratliff [4,5], Kent L. Nastiuk[3,8], Susan E. Crawford[1] & Simon W. Hayward [1✉]

Autoimmune (AI) diseases can affect many organs; however, the prostate has not been considered to be a primary target of these systemic inflammatory processes. Here, we utilize medical record data, patient samples, and in vivo models to evaluate the impact of inflammation, as seen in AI diseases, on prostate tissue. Human and mouse tissues are used to examine whether systemic targeting of inflammation limits prostatic inflammation and hyperplasia. Evaluation of 112,152 medical records indicates that benign prostatic hyperplasia (BPH) prevalence is significantly higher among patients with AI diseases. Furthermore, treating these patients with tumor necrosis factor (TNF)-antagonists significantly decreases BPH incidence. Single-cell RNA-seq and in vitro assays suggest that macrophage-derived TNF stimulates BPH-derived fibroblast proliferation. TNF blockade significantly reduces epithelial hyperplasia, NFκB activation, and macrophage-mediated inflammation within prostate tissues. Together, these studies show that patients with AI diseases have a heightened susceptibility to BPH and that reducing inflammation with a therapeutic agent can suppress BPH.

[1] Department of Surgery, NorthShore University HealthSystem, an Academic Affiliate of the University of Chicago Pritzker School of Medicine, Evanston, IL 60201, USA. [2] Department of Cancer Biology, Meharry Medical College, Nashville, TN 37208, USA. [3] Department of Cancer Genetics and Genomics, Roswell Park Comprehensive Cancer Center, Buffalo, NY 14263, USA. [4] Department of Comparative Pathobiology, Purdue University, West Lafayette, IN 47907, USA. [5] Purdue Center for Cancer Research, Purdue University, West Lafayette, IN 47907, USA. [6] Biostatistics and Research Informatics, NorthShore University HealthSystem, Evanston, IL 60201, USA. [7] Department of Medicinal Chemistry and Molecular Pharmacology, Purdue University, West Lafayette, IN 47907, USA. [8] Department of Urology, Roswell Park Comprehensive Cancer Center, Buffalo, NY 14263, USA. [9] Present address: Department of Quantitative Health Sciences, Lerner Research Institute, Cleveland Clinic, Cleveland, OH 44195, USA. [10] Present address: Department of Nephro-Urologic Surgery and Andrology, Mie University Graduate School of Medicine, Mie, Japan. [11] Present address: GW Cancer Center, The George Washington University, Washington, DC 20052, USA. ✉email: shayward@northshore.org

The vast majority of men develop histological benign prostatic hyperplasia (BPH) and nearly half exhibit moderate to severe clinical symptoms before age 80[1,2]. The precise mechanisms underlying the pathogenesis of BPH and how it contributes to lower urinary tract symptoms (LUTS) are not well understood. The only definitive risk factors for developing BPH are male sex and increasing age, but BPH has been linked to decreased systemic androgen/estrogen ratios, obesity, type 2 diabetes, metabolic syndrome, and inflammation[3–8]. There is a paucity of studies investigating possible genetic determinants of BPH, although, more recently, there has been greater interest in this area[9–11]. The medical therapeutic options used for men with LUTS related to BPH are limited (e.g. alpha-adrenergic antagonists and 5α-reductase inhibitors [5ARIs][12]) and have not changed significantly for two decades. Due to therapeutic resistance or disease progression, over 100,000 men undergo surgical procedures for BPH each year in the United States[13]. A non-surgical, targeted approach for medical treatment of these BPH cases is needed.

Chronic inflammation in the prostate can contribute to prostatic hyperplasia, fibrosis, and failure to respond to therapy in BPH[12,14,15]. CD45+ leukocytes are known to comprise a significant percentage of cells in BPH tissues, with macrophages and T cells as major populations[16]. CD68+ macrophages accumulate in BPH tissues and aid in stromal cell proliferation[17]. Whether BPH is linked to an inflammatory process is not known, although BPH has been suggested to have characteristics of autoimmune (AI) inflammatory conditions[18,19]. An association between AI and BPH is consistent with our observations of an inflammatory gene expression signature including activation of AP-1 stress factors associated with severely symptomatic BPH that was refractory to medical therapy and required surgical intervention[20]. BPH is associated with a number of common pro-inflammatory comorbidities[4–7], suggesting that the systemic environment may promote hyperplasia and/or exacerbate symptoms. Although there is no perfect animal model to study human BPH, some, such as non-obese diabetic (NOD) mice, recapitulate the association of chronic prostatic inflammation with prostatic hyperplasia, similar to the observations in human disease[21].

AI conditions such as rheumatoid arthritis (RA) and systemic lupus erythematosus, which are associated with systemic inflammation, can involve multiple organ sites; however, the prostate has not been recognized as a target organ of these inflammatory processes. Interestingly, AI diseases have similar comorbidities to BPH, including obesity, type 2 diabetes, and metabolic syndrome[22–25], and some AI conditions are also comorbidities of other AI diseases, such as psoriasis and inflammatory bowel disease[26]. AI diseases are significantly more prevalent in women compared to men, potentially highlighting an immunosuppressive function of androgens in protecting against AI diseases[27–29].

The earliest and most widely used therapeutics for targeting a specific biological pathway in AI conditions are tumor necrosis factor (TNF)-antagonists, which limit the inflammatory properties of this cytokine[30]. Recent single-cell mRNA-sequencing (scRNA-seq) studies in AI diseases have identified a variety of immune cell populations including monocytes/macrophages, T cells, and B cells in diseased tissues[31–33]. Notably, ligand-receptor pair interaction analyses highlight the impact of macrophage-secreted TNF on stromal cells in Crohn's disease[33]. It is clear that TNF ligand-receptor signaling is important within AI diseases, but the contribution of this cytokine to the development or progression of BPH in the AI patient population has not been studied.

Here, we investigate whether BPH prevalence is increased among men with AI conditions and test whether therapies for AI diseases can reduce BPH diagnoses or suppress prostatic inflammation in BPH. Patient medical records and human prostate tissues are used to support the repurposing of approved therapeutics. The results suggest that TNF-antagonists may be viable therapeutics to reduce BPH incidence in patients with AI diseases and that these drugs decrease localized inflammation within the prostate.

## Results

**BPH prevalence is elevated in specific autoimmune diseases.** An institutional review board (IRB)-approved retrospective evaluation of the NorthShore University HealthSystem Enterprise Data Warehouse was conducted to determine whether patients with autoimmune (AI) disease had an elevated risk of BPH diagnosis. The codes and medications used for data collection are indicated in Supplementary Tables 1–3. Male patients over the age of 40 who had a NorthShore office visit between 01/01/2010-12/31/2012 were included (n = 112,152; Supplementary Table 4). Patients with a diagnosis of prostate cancer were excluded from evaluation. Records were searched for diagnoses of BPH and a range of AI conditions, most commonly psoriasis, RA, and ulcerative colitis. The majority of patients were non-Hispanic Caucasians (Supplementary Tables 5, 6). The cohort included 101,383 (90.4%) men with no history of AI and 10,769 (9.6%) men with a diagnosis of one or more AI conditions (Fig. 1a; Supplementary Data 1). To determine whether treatment for AI diseases affected BPH diagnosis in this population, patients with an AI disease diagnosis were further divided into those who had a diagnosis of AI disease prior to a BPH diagnosis or those who had a diagnosis of AI disease after a BPH diagnosis (Fig. 1a). Results from a chi-square test indicated that the baseline prevalence of BPH was 20.3% in patients with no history of AI disease, but patients with diagnosed AI conditions had a significantly increased BPH prevalence of 30.6% (p < 0.001; Fig. 1b, c; Supplementary Table 7). The most marked increases in BPH prevalence were associated with RA (38.0%), type 1 diabetes (32.0%), lupus (30.7%), ulcerative colitis (30.2%), and Crohn's disease (27.4%), while other diseases, such as multiple sclerosis (21.6%), showed little change from baseline (Fig. 1d; Supplementary Table 7).

In the subset of men who were diagnosed with their AI condition prior to BPH diagnosis, the BPH incidence was 19.4% via chi-square test, similar to the baseline BPH prevalence (p = 0.037; Fig. 1e; Supplementary Table 7). This suggests that treatment of AI disorders significantly diminishes subsequent BPH diagnoses, although there were some disease-specific variations in BPH incidence with treatment (Fig. 1f; Supplementary Table 7). Nearly all AI conditions in this group had a significantly lower incidence of BPH compared to the baseline in patients without AI disease using chi-square tests, with the exception of RA patients who remained at a significantly elevated rate of BPH diagnosis compared to control patients (23.2%; p = 0.007). However, this BPH incidence was reduced from the BPH prevalence in all RA patients (38.0%; Fig. 1d, f).

**AI disease patients treated with TNF-antagonists have decreased BPH diagnoses.** Using the complete patient population from Fig. 1, patient age and the use of common therapeutics, specifically methotrexate and TNF-antagonists, were modeled as predictors of BPH diagnosis using multivariable logistic regression. Multivariable logistic regression models were adjusted for age, race, ethnicity, and body mass index (BMI) parameters. Subjects over the age of 60 were significantly more likely (odds ratio=8.18; p < 0.001) to have BPH than those under age 60 (Table 1). This analysis indicated that treatment with specific

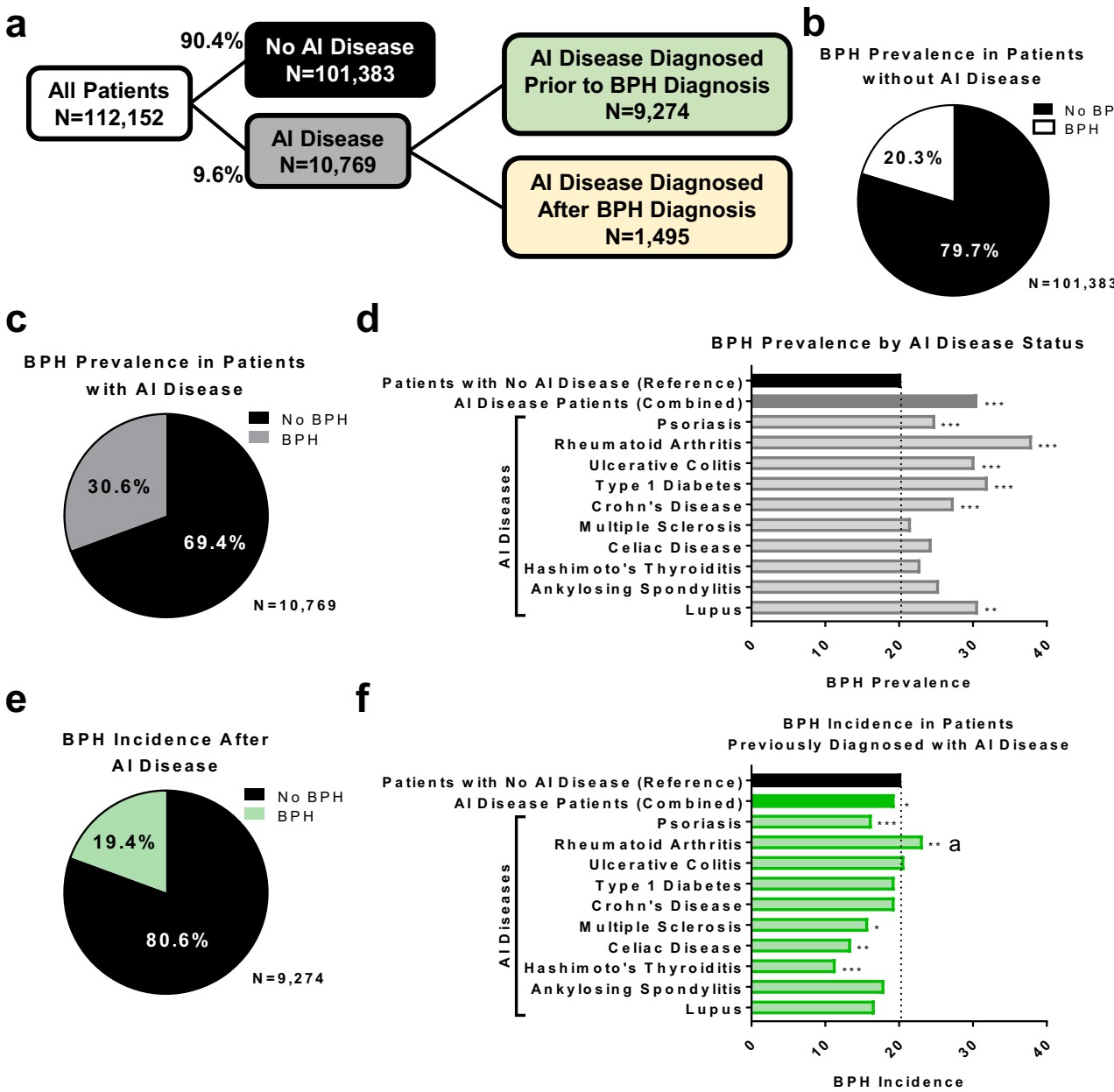

**Fig. 1 Men with autoimmune disease have increased BPH prevalence.** Chi-square tests were utilized to compare the proportion of BPH diagnoses in men with an AI condition versus men with no AI condition. Colors indicate categories of patients, where black = patients without AI disease, gray = patients with AI disease, green = patients diagnosed with AI disease prior to BPH diagnosis, and yellow = patients diagnosed with AI disease after BPH diagnosis. **a** Flow chart indicating the breakdown of patients into groups based on the presence of AI disease diagnosis (9.6% with and 90.4% without AI disease diagnosis). Patients with AI disease diagnosis were further separated into groups based on whether AI disease diagnosis occurred prior to or after BPH diagnosis. **b** BPH prevalence in patients without AI disease is 20.3%. **c** BPH prevalence in patients with AI disease is 30.6%. **d** Graph indicates the significant increase in BPH prevalence in patients with different AI diseases compared to patients without AI disease using chi-square tests. **e** The BPH incidence in patients previously diagnosed with AI conditions, when patients may have been treated for these conditions, is 19.4%. **f** Chi-square tests indicate the significant changes in BPH incidence in patients diagnosed with different AI diseases prior to BPH diagnosis compared to the baseline BPH prevalence of 20.3%. [a] indicates a significantly higher BPH incidence than the 20.3% reference, although this is decreased from 38.0% prevalence in all RA patients (d). *$p < 0.05$, **$p < 0.01$, and ***$p < 0.001$.

therapeutics, namely TNF-antagonists but not methotrexate, significantly decreased (odds ratio = 0.79; $p = 0.033$) the likelihood of a BPH diagnosis compared to patients not taking these drugs (Table 1). In the AI disease patient population, the vast majority of patients were taking either methotrexate or TNF-antagonists, with differences depending on the specific AI disease (Supplementary Table 8). Further evaluation of these drugs using chi-square or Fisher's exact tests in patients with AI disease

indicates that methotrexate use significantly increases BPH diagnosis ($p < 0.0001$) while the use of TNF-antagonists significantly decreases BPH diagnosis ($p = 0.006$) compared to patients without AI disease (Supplementary Table 9). Since patients with AI disease have an elevated BPH prevalence compared to patients without AI disease, a similar comparison of the BPH incidence in patients taking methotrexate or TNF-antagonists to that of patients not taking medication within

**Table 1 Modeling predictors of BPH diagnosis indicate that treatment of patients with TNF-antagonists is protective for BPH.**

| Predictors of BPH diagnosis | N (%) | Odds ratio | 95% CI | p-value |
|---|---|---|---|---|
| Age 60+ Medication | 52,393 (46.7) | 8.18 | 7.89-8.49 | **<0.001** |
| Methotrexate | 708 (0.63) | 1.58 | 1.32-1.90 | **<0.001** |
| TNFα-antagonists | 730 (0.65) | 0.79 | 0.64-0.98 | **0.033** |

Multivariable logistic regression models were constructed to identify predictors of BPH diagnosis. The models were adjusted for age, race, ethnicity, and body mass index (BMI). Predictors were determined a priori: age over 60 years, methotrexate, and TNF-antagonists. These studies included all 112,152 patients in the cohort. Significant p-values are bolded.

each AI disease was performed. Again, these data indicate that methotrexate significantly increases BPH incidence ($p < 0.0001$) while TNF-antagonists significantly decrease BPH incidence ($p = 0.0314$) compared to AI patients not taking medication (Supplementary Table 9). Conducting the comparison of TNF-antagonist treatment versus no medication for the patients within AI disease subcategories does not reach significance, likely due to the small sample sizes (Supplementary Table 9). Instead, separating related AI diseases such as Crohn's disease and ulcerative colitis, both forms of inflammatory bowel disease (IBD), followed by Kaplan–Meier analysis indicates that TNF-antagonist treatment results in the lowest probability of BPH diagnosis in patients with either of these related AI diseases ($p = 0.0054$; Supplementary Fig. 1), in all AI diseases other than IBD ($p < 0.0001$; Supplementary Fig. 2), as well as in all AI diseases combined ($p < 0.0001$; Supplementary Fig. 3).

While TNF-antagonists are currently used to treat a number of AI conditions, a limited number of studies have suggested that treatment with these drugs can actually increase incidence of a subset of AI conditions, including Crohn's disease and ulcerative colitis[34,35]. To determine whether the use of TNF-antagonists in this cohort affected the frequency of diagnosis of IBD, the rate of subsequent ulcerative colitis or Crohn's disease diagnosis after medication use for other AI conditions was evaluated using Fisher's exact test. The analysis showed that AI disease patients were most likely to develop subsequent IBD if they were not treated with any medication ($p < 0.0001$; Supplementary Tables 10, 11). However, post-hoc analysis using Bonferroni correction indicated that TNF-antagonist treatment significantly increased the risk of ulcerative colitis ($p < 0.0001$), but not Crohn's disease ($p = \text{n.s.}$), compared to other medication categories (Supplementary Tables 10, 11).

**scRNA-seq of BPH leukocytes implicate T cells and macrophages as sources of TNF.** The decrease in BPH incidence after treatment with systemic anti-inflammatory agents that target TNF suggests a function for inflammation in BPH pathogenesis. Therefore, scRNA-seq studies were pursued to characterize inflammatory cells within the prostatic transition zone. Tissues were obtained from patients with informed consent, per the IRB-approved NorthShore Urologic Disease Biorepository. As prostate size increases with BPH, a t test indicates that immune cell density also increases ($p = 0.0021$; Fig. 2a, b). To evaluate the TNF-producing and TNF-responding inflammatory cell types within human BPH tissues, scRNA-seq analysis on CD45$^+$ cells was conducted in two groups (Supplementary Fig. 4). The first group represented limited prostatic growth and included cells from the transition zone of smaller (<40 g, $n = 10$) prostates, whereas the second group represented

prostatic enlargement using cells from the transition zone of large (>90 g, $n = 4$) prostates (Fig. 2c). As shown using $t$ tests, prostate tissues were isolated from age-matched ($p = 0.32$) and BMI-matched ($p = 0.13$) patients (Supplementary Fig. 5a, b). Patients with larger prostates exhibited significantly higher International Prostate Symptom Scores (IPSS) compared to patients with smaller prostates, completed via $t$ test ($p = 0.0007$; Supplementary Fig. 5c, d).

Unsupervised clustering analysis of scRNA-seq data allowed for the identification of various cell populations using differential gene expression and cell surface protein expression using CITE-seq[36]. T cells (CD3$^+$ and CD4$^+$ or CD8$^+$) and macrophages (CD11b$^+$) comprised the dominant immune cell populations but B cells (CD19$^+$), mast cells, NK cells, and plasma cells were also identified (Fig. 2d, e; Supplementary Fig. 5e). Although no significant differences in overall immune subpopulations were noted when comparing large versus small tissues or from individual patients (Supplementary Fig. 5f, g), several of the most significantly altered pathways between large and small samples included pathways related to AI conditions (Supplementary Table 12). The percentage of CD11b$^+$ myeloid cells, CD19$^+$ B cells, and CD4$^+$ or CD8$^+$ T cells from the original digested samples by flow cytometry analysis were correlated with the percentage of these cell types based on cluster identification in the scRNA-seq analysis. Linear regression identified significant correlations for myeloid cells, B cells, and CD8$^+$ T cells, indicating that the cell types identified by scRNA-seq were representative of the CD45$^+$ population in the digested human tissues (Supplementary Fig. 6). Cells with the highest expression of *TNF* were within the T cell and macrophage compartments, and macrophages (highlighted as CD68$^+$) also expressed high levels of TNF receptors 1 and 2, *TNFRSF1A* and *TNFRSF1B*, respectively (Fig. 2f, Supplementary Fig. 4). The Wilcoxon rank sum test corrected for multiple testing using the Benjamini-Hochberg method indicated that macrophage cluster 0 had significantly higher expression of *TNF* (fold-change=1.364; $p < 0.0001$) and *TNFRSF1A* (fold-change=1.325; $p < 0.0001$), while cluster 9 had significantly higher expression of *TNFRSF1A* (fold-change=1.706; $p < 0.0001$), compared to all other clusters (Supplementary Fig. 7).

While BPH macrophages express genes indicating a likely response to TNF within the immune cell compartment, it is also important to note that this remains the top cell type expressing TNF receptor genes even when evaluating all prostate cells by scRNA-seq. In an evaluation of all, unsorted cells from five human BPH tissues after simple prostatectomy, all major cell populations were identified from each patient (Supplementary Fig. 8a, b) and elevated expression of *TNF* and *TNFRSF1B* was restricted to immune cell populations (Supplementary Fig. 8C–F). *TNFRSF1B* was specifically elevated in CD68-expressing macrophages, while *TNFRSF1A* had no detectable transcripts after sequencing. Taken together, BPH macrophages had the greatest potential for expression of genes related to secretion and response to TNF, so we focused on these cells.

**TNF-antagonist treatment causes prostatic regression in Pb-PRL mice.** To assess the impact of TNF-antagonists on BPH and prostatic inflammation, two mouse models of prostatic enlargement were utilized: a transgenic model with prostate-specific expression of the hormone prolactin (probasin-prolactin [Pb-PRL]) that exhibits prostatic enlargement associated with extensive interstitial inflammation[37,38] as well as a spontaneous model of autoimmune inflammation-associated prostatic hyperplastic growth (the NOD mouse)[21].

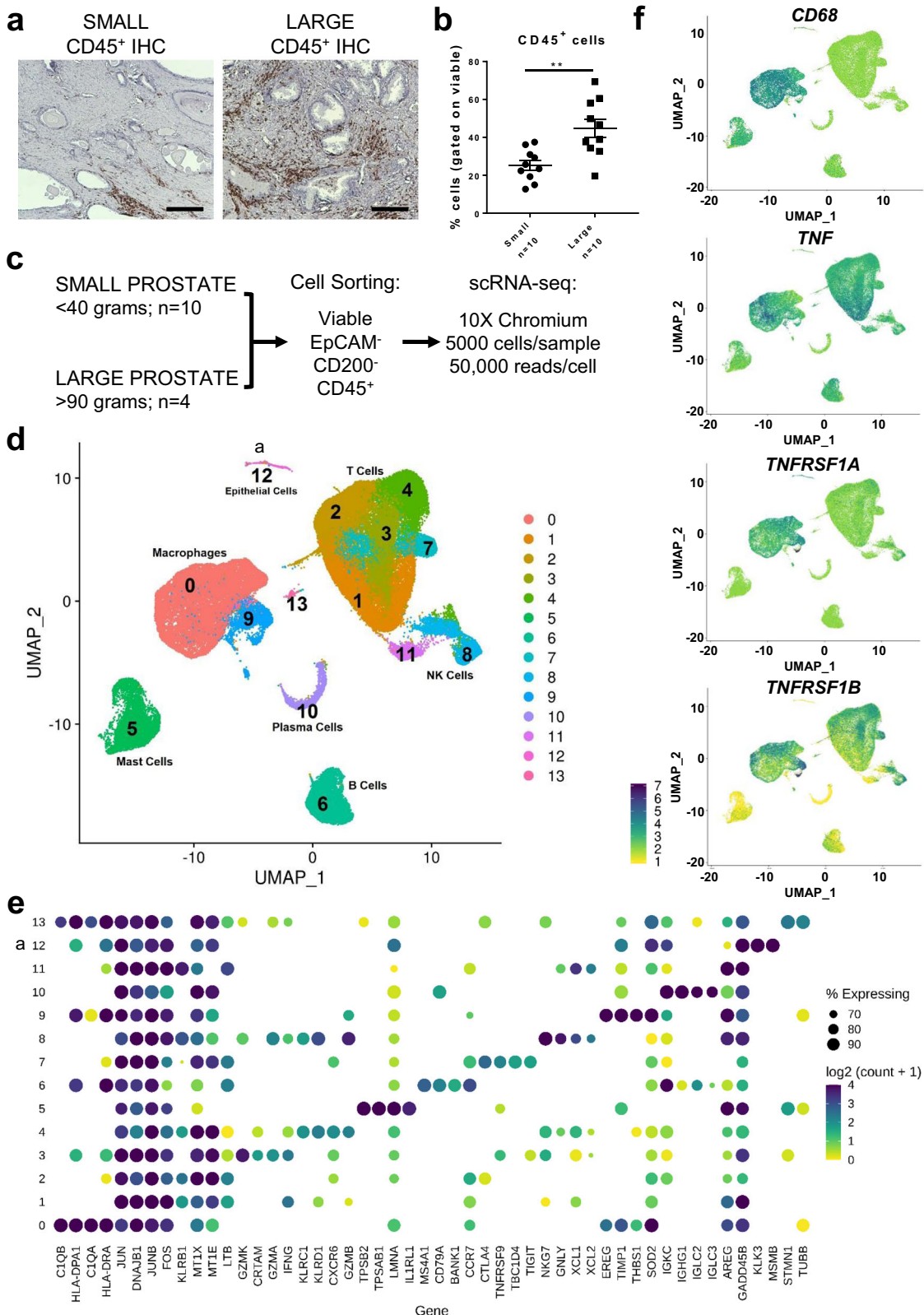

Pb-PRL mice have been shown to develop prostatic hyperplasia with associated interstitial inflammation[37,38]. Aged Pb-PRL mice (20–22 months) were treated with 4 mg/kg TNF-antagonist etanercept or PBS control twice weekly for 12 weeks ($n = 6$ in control group, $n = 5$ in treated group). The dosing regimen of etanercept was sufficient to block intraprostatic TNF signaling[39]. Ventral prostate volume was monitored by ultrasound every four weeks using high-resolution, high-frequency ultrasound[40]. Linear model analysis indicated that ventral prostate volume trended downward after eight weeks and was reduced over 30% after 12 weeks of treatment ($p = 0.0365$; Fig. 3a). Histological evaluation of prostate tissues following etanercept treatment indicated significantly diminished epithelial cell proliferation via Ki67 immunohistochemistry (IHC) staining compared to control

**Fig. 2 Analysis of CD45$^+$ cells from human BPH tissues indicate macrophages express high levels of TNF and TNF receptors. a** CD45 IHC in human prostate transition zone from small or large prostates ($n = 4$ patients per group). Staining was confirmed in two independent experiments. Brown color indicates positive CD45 staining and scale bars = 500 µm. **b** Human prostate transition zone from small or large prostates ($n = 10$ patients per group) were digested, stained, and analyzed by flow cytometry. Graph indicates % CD45$^+$ cells, gated on viable cells. A significant difference in %CD45$^+$ cells (**$p = 0.0021$) using a two-tailed $t$ test. Error bars represent the mean ± SEM. Source data are provided as a Source Data file. **c** Schematic representing the setup for scRNA-seq studies of human BPH leukocytes (CD45$^+$ cells). A total of 10 small and 4 large prostate transition zone tissues were digested and CD45$^+$EpCAM$^-$CD200$^-$ cells sorted by FACS. scRNA-seq was conducted using the 10X Chromium system, aiming for 5000 cells/sample at a depth of 50,000 reads/cell. **d–f** scRNA-seq of BPH associated leukocytes. **d** Uniform manifold approximation and projection (UMAP) plot of 69,850 individual cells from 14 patient samples, demonstrating dominant T cell and macrophage populations. Each color indicates a unique cell cluster. **e** Dot plot of the top 4 marker genes from each cluster ranked by fold-change. Gene names are shown on the $x$-axis and clusters on the $y$-axis. The size of the dots corresponds to the percentage of cells in a given cluster that express the marker gene. The color of the dots represents the mean log$_2$(counts + 1) of each gene in the corresponding cluster. **f** Feature plots highlighting gene expression of *CD68*, *TNF*, *TNFRSF1A* (TNFR1), and *TNFRSF1B* (TNFR2), where the blue color indicates elevated expression of the indicated gene. $^a$ indicates a small contaminating epithelial population as cluster 12.

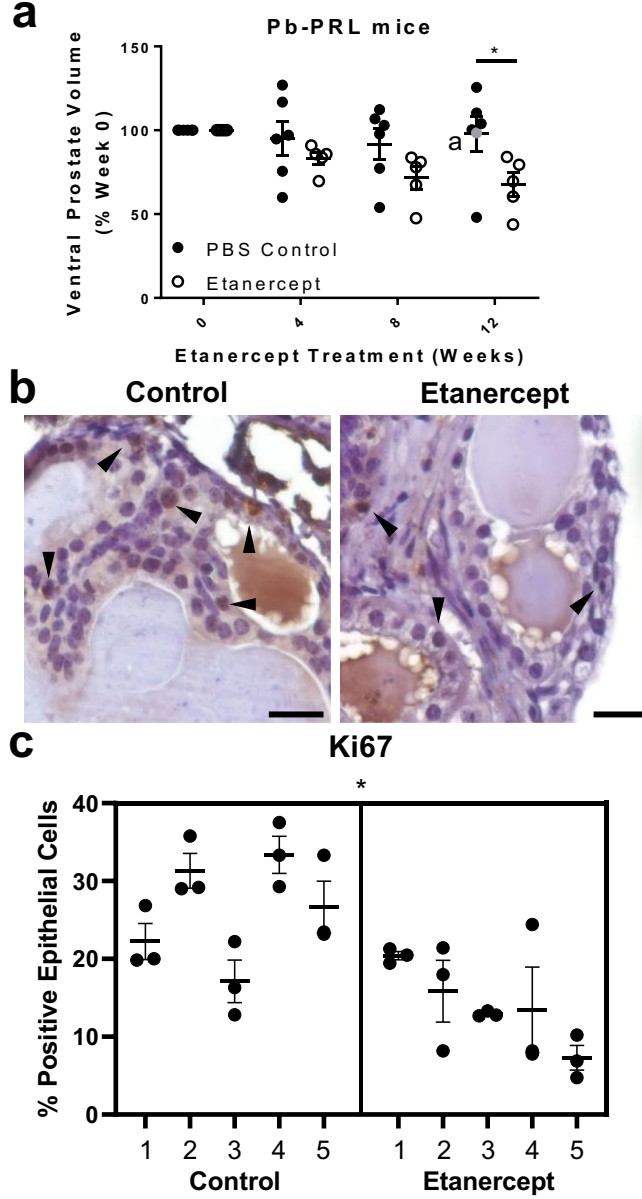

**Fig. 3 TNF-antagonist treatment reduces prostate size and epithelial proliferation in Pb-PRL mice.** Pb-PRL mice (20-22 months) were treated with 4 mg/kg etanercept or PBS vehicle for 12 weeks. **a** Volume of ventral prostate by ultrasound every four weeks during the 12-week treatment with etanercept or PBS vehicle control. Measurements are normalized to pre-treatment volume (151 ± 10 mm$^3$), and the plot indicates the mean ± SEM. One control mouse was removed from the 12-week evaluation, but the data point is included for reference and indicated by a gray point ●. A two-sided linear mixed model analysis showed a significant difference based on treatment ($p < 0.0001$) and Bonferroni correction determined a reduction in ventral prostate volume compared to PBS-treated mice at 12 weeks (*$p = 0.0365$; $n = 5$ per group). Source data are provided as a Source Data file. **b** Representative images of Ki67 staining in control or etanercept-treated mice, where brown color indicates positive staining. **c** Quantitation of IHC staining for epithelial Ki67 in control or etanercept-treated mice ($n = 5$ animals per group), indicated as the percent Ki67 positive epithelial cells per field of view. Data indicate the mean ± SEM of the percent positive cells in three prostate tissue fields for each animal. Comparison of control and etanercept-treated groups determined a significant difference (*$p = 0.0105$) using a two-tailed nested $t$ test. Source data are provided as a Source Data file. Scale bars = 20 µm. $^a$ indicates an animal excluded from statistical analysis.

epithelial NFκB activation via phospho-p65 staining ($p = 0.8748$ and $p = 0.8586$, respectively; Supplementary Fig. 9).

**TNF-antagonists reduce prostate hyperplasia, inflammation, and NFκB in NOD mice.** We recently reported that NOD mice have inflammation-associated prostatic hyperplasia[21]. Therefore, these mice were used as a model to understand the consequence of TNF blockade on hyperplastic expansion. Animals were treated with 4 mg/kg etanercept or vehicle control twice weekly for five weeks, starting at 25 weeks, as indicated in Fig. 4a. Diabetic status was evaluated at the time of first and last treatment, although our previously published work indicated that regions of inflammation, rather than diabetic status, primarily impacts prostatic hyperplasia in NOD mice[21]. Therefore, diabetic and non-diabetic animals were pooled for analysis of treatment groups. Quantitation of Ki67 IHC staining using a nested $t$ test indicated a reduction in epithelial proliferation in etanercept-treated versus control mice ($p = 0.0002$; Fig. 4b, c), consistent with the Pb-PRL model even though no change in ventral prostate weight was determined (Supplementary Fig. 10a). A similar evaluation of prostatic macrophages via F4/80 staining demonstrated a significant decrease in the percentage of macrophages among total immune cells in etanercept-treated versus control mice ($p = 0.0002$; Fig. 4d, e). Furthermore, etanercept treatment caused a reduction in epithelial NFκB activity, as demonstrated by a significant reduction in epithelial phospho-p65 staining by nested

tissues using a nested $t$ test ($p = 0.0105$; Fig. 3b, c). Similar evaluation of Pb-PRL prostate tissues of etanercept-treated versus control mice did not yield significant differences for either macrophage infiltration as a percentage of immune cells or

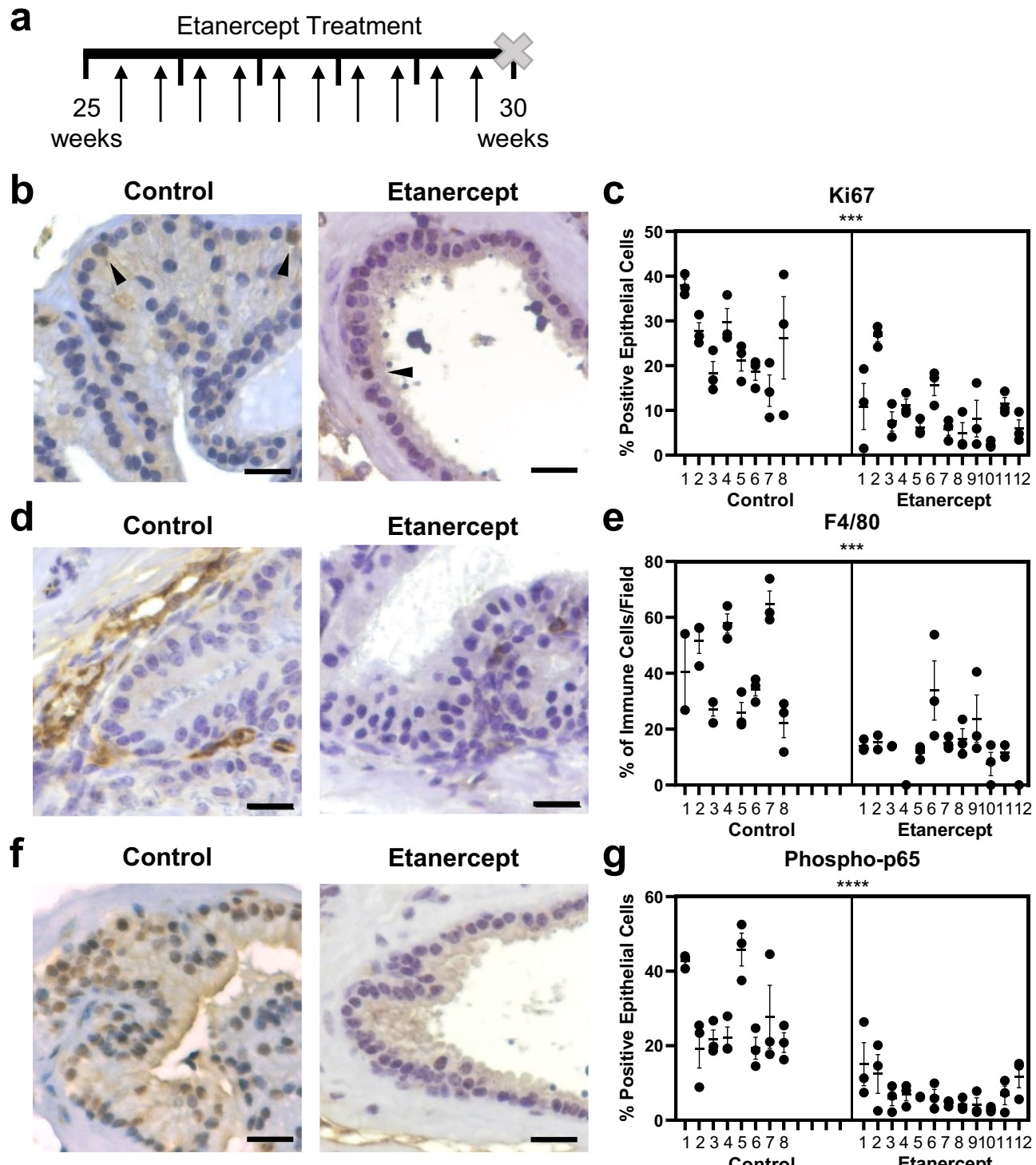

**Fig. 4 TNF-antagonist treatment reduces prostatic epithelial proliferation, macrophage infiltration, and NFκB activity in NOD mice. a** Diagram representing the timeline of treatment in NOD mice. Mice were treated twice per week for 5 weeks with 4 mg/kg etanercept or PBS vehicle, indicated with black arrows. At the end of the 5-week treatment period, tissues were harvested for analysis. **b** Representative images of Ki67 staining in control or etanercept-treated mice, where brown color indicates positive staining. **c** Graph indicating the quantitative summary of Ki67 IHC as the percent of positive epithelial cells per field (***$p = 0.0002$). **d** Representative images of F4/80$^+$ staining in control or etanercept-treated mice, where brown color indicates positive staining. **e** Quantitative summary of IHC staining for F4/80$^+$ cells, represented as the portion of all immune cells in each field (***$p = 0.0002$). **f** Representative images of phospho-p65 staining in control or etanercept-treated mice, where brown color indicates positive staining. **g** Data presented indicate the percentage of phospho-p65 positive epithelial cells counted per field of view (****$p < 0.0001$). **c, e, g** Data indicate the mean ± SEM of the percent positive cells in the indicated number of prostate tissue fields for each animal ($n = 8$ for control-treated and n = 12 for etanercept-treated). Comparison of control and etanercept-treated groups for statistical purposes was conducted using a two-tailed nested *t* test and asterisks indicate the significance of the treatment. Source data are provided as a Source Data file. Scale bars = 20 μm.

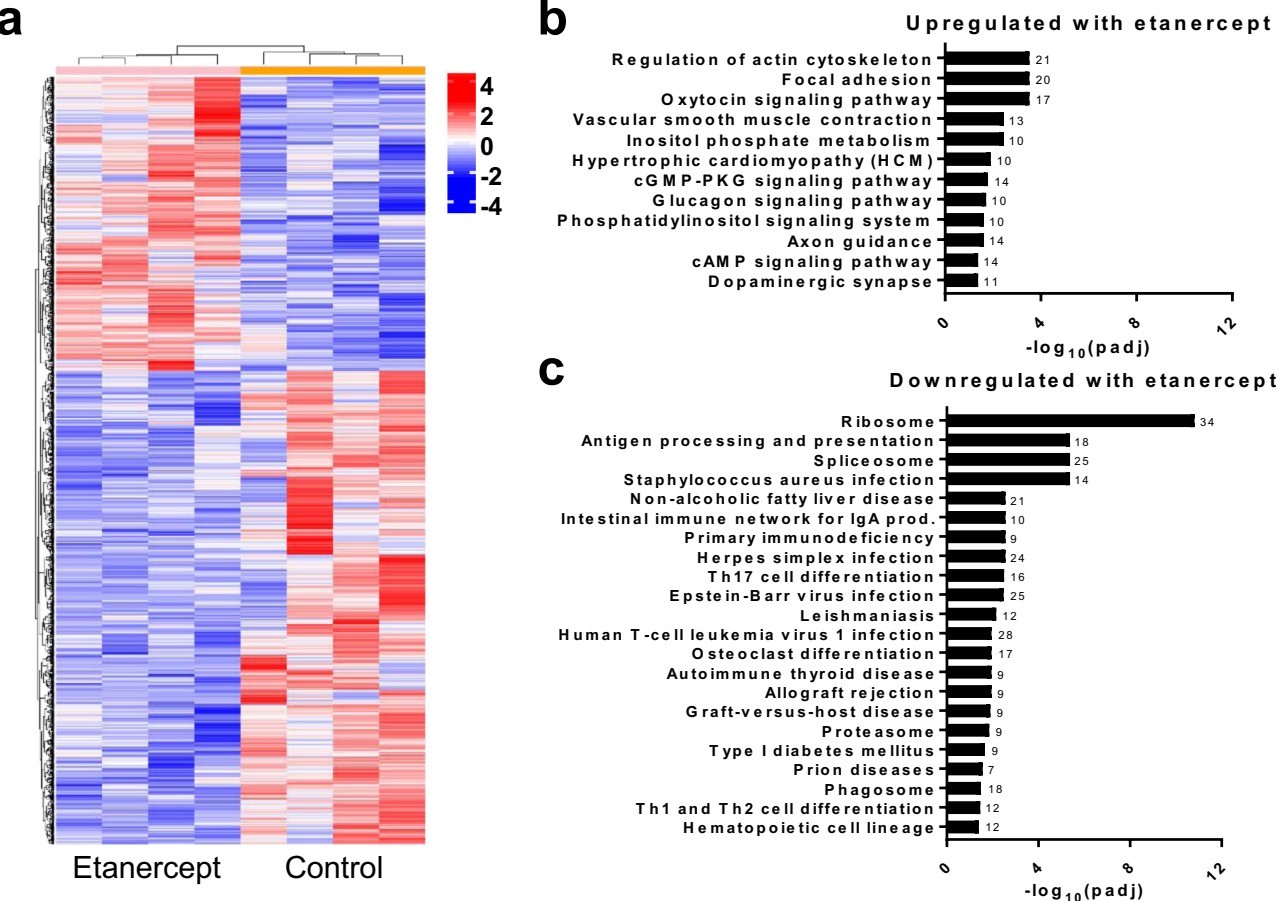

**Fig. 5 Etanercept treatment induces pathway alterations in NOD prostate tissues.** Bulk RNA-seq was conducted on the prostate tissues from control and etanercept-treated NOD mice (n = 4 per group) after five weeks of treatment. **a** Hierarchical clustering of samples based on DE genes indicates separate clustering of the two treatment groups. Clustering analysis is carried out by $\log_2$(FPKM + 1) of union DE genes. Red color indicates upregulated genes and blue color indicates downregulated genes. **b**, **c** KEGG enrichment analysis using a one-tailed Fisher's exact test identified significantly upregulated (**b**) and downregulated (**c**) pathways in response to etanercept treatment (adjusted p-value< 0.05). Numbers listed with each bar indicate the number of altered genes within each pathway. Source data are provided as a Source Data file.

t test (p < 0.0001; Fig. 4f, g). These data provide an association between the inflammatory cytokine TNF with epithelial proliferation and NFκB signaling in the prostate, even though anti-etanercept autoantibodies were identified to be present in NOD mice after 5 weeks of etanercept treatment (Supplementary Fig. 10b). The differences between the Pb-PRL and NOD models could reflect differences in the age at or duration of treatment as well as the nature of the model, since NOD is a spontaneous autoimmune inflammatory model and Pb-PRL is an androgen-driven transgenic with unknown contributions of inflammatory status on androgenic stimulation.

**RNA-seq identifies pathway alterations in tissues treated with etanercept.** To evaluate the molecular impact of etanercept treatment in the prostate, bulk RNA-seq was conducted on control and etanercept-treated NOD prostate tissues to identify putative pathways involved in these processes. Hierarchical clustering successfully separated these groups based on differential gene expression (Fig. 5a). As anticipated, etanercept treatment led to downregulation of TNF signaling pathway genes, specifically downstream of TNFRSF1B (Supplementary Fig. 11). Significantly upregulated and downregulated pathways between etanercept-treated and control mice were determined using KEGG enrichment analysis with Fisher's exact test (Fig. 5b, c). Notable pathways upregulated by etanercept treatment include

focal adhesion (p = 0.0003) and regulation of actin cytoskeleton (p = 0.0003; Supplementary Fig. 12), with other pathways suggesting muscle contraction (Fig. 5b). Notable downregulated pathways in treated tissues primarily relate to inflammatory signaling, such as antigen processing and presentation (p < 0.0001; Supplementary Fig. 13) and autoimmune thyroid disease (p = 0.0007; Fig. 5c). These data suggest that TNF blockade dramatically reduces inflammatory signaling but activates alternative pathways of other stromal cell types in the prostate.

**TNF stimulates fibroblast but not epithelial cell proliferation in vitro.** Fibroblasts are an essential stromal component of the prostate microenvironment during organogenesis and in BPH pathogenesis[41]. Since TNF-antagonist treatment significantly reduces epithelial proliferation in two independent mouse models, we tested whether TNF directly influences epithelial and fibroblast cell growth in vitro. Importantly, treatment of two human benign prostatic epithelial cell lines elicited no changes in cell growth response to 1 or 10 ng/mL TNF (Supplementary Fig. 14a, b). In contrast, similar treatment with TNF in a human benign prostatic stromal cell line enhanced cell proliferation significantly, as determined by two-way ANOVA (p < 0.0001; Supplementary Fig. 14c). Similarly, TNF directly stimulated the proliferation of primary human prostatic fibroblast cultures from four different patients (p < 0.0001; Fig. 6a). Furthermore,

treatment of primary BPH fibroblasts with conditioned medium from M1- or M2-polarized THP-1 cells demonstrated that macrophage-secreted factors stimulate fibroblast growth (Fig. 6b). The addition of a TNF neutralizing antibody indicated that macrophage-stimulated growth can be TNF-dependent, as indicated by two-way ANOVA ($p = 0.0011$ and $p = 0.0148$ for M1 and M2 in patient 376, respectively; $p < 0.0001$ and $p = 0.0004$ for M1 and M2 in patient 1579, respectively), although this may be patient-specific (Fig. 6b). Macrophage conditioned medium did not alter epithelial cell proliferation (Supplementary Fig. 15). Since epithelial cell proliferation was not directly affected by TNF treatment in vitro, these data are consistent with the long-standing idea that prostate expansion in BPH has a significant stromal input in vivo.

**TNF-antagonists decrease prostatic inflammation and proliferation in patients.** To better understand the impact of TNF-antagonist treatment on human prostate tissues, we performed a retrospective study using deidentified human prostate tissues collected through the NorthShore Urologic Disease Biorepository. The biorepository provided transition zone tissues from patients taking TNF-antagonists who also underwent surgery for prostatic diseases ($n = 5$). Age- and BMI-matched patient samples were used as controls ($n = 5$; Supplementary Fig. 16a, b). All of the patients in both the treated and control groups had a robotic-assisted laparoscopic prostatectomy (RALP) due to cancer, but also had a clinical diagnosis of BPH. No significant changes in prostate volumes or IPSS were observed in treated patients compared to controls using a t test (Supplementary Fig. 16c, d). Benign tissue regions from patients treated with TNF-antagonists had significantly less epithelial Ki67 staining compared to tissues from control patients, using a nested t test ($p = 0.0123$; Fig. 7a, b). In general, the overall level of inflammation appeared to be lower in tissues from treated patients, and quantitative evaluation of CD68$^+$ cells via IHC and nested t test indicated that TNF-antagonist treatment significantly reduced the percentage of CD68$^+$ macrophages out of total immune cells compared to controls ($p = 0.0148$; Fig. 7c, d). Tissues from TNF-antagonist treated patients also had significantly diminished epithelial NFκB activity compared to tissues from control patients, as demonstrated by phospho-p65 staining and analysis with nested t test ($p = 0.0172$; Fig. 7e, f). Together, these data suggest that epithelial hyperplasia in BPH may be promoted by inflammation-derived TNF and that this could be abrogated by systemic treatment with TNF-antagonists.

## Discussion

The results from these studies indicate that TNF-antagonists, a well-tolerated class of drugs commonly used to treat AI disease, alter the pathogenesis of BPH. These agents, but not methotrexate, reduce BPH incidence in patients with AI diseases. In animal models, TNF-antagonists reduce both the establishment of epithelial hyperplasia and, in aged mice, prostate size. A number of trials using NSAIDs to relieve BPH symptoms have shown some positive effects, but limited long-term efficacy[42,43]. The present data strongly suggest that appropriate targeting of specific molecular signals may be more clinically effective than broad-spectrum agents, but the utility of NSAIDs in combination with TNF-antagonists requires further study. There are many immunomodulatory drugs in development or already being prescribed for AI diseases that could be tested for their activity in BPH.

These studies strengthen the link between prostate inflammation and the development of BPH. NFκB activation has been shown to be increased in BPH samples progressing to disease-specific surgery and was associated with increased expression of 5α-reductase 2, increased androgen receptor expression, and increased cellular proliferation in vitro[44,45], and TNF may drive this paradigm. Inflammation can directly affect molecular pathways associated with growth and resistance to 5ARI therapies[44,45]. Furthermore, natural remedies such as saw palmetto and beta-sitosterol have also been suggested to reduce NFκB activity[46,47], indicating that NFκB activation may be involved in driving and/or supporting hyperplastic growth.

The immunosuppressive function of androgens has been reported in numerous studies[29]. The contribution(s) of the androgen receptor in BPH is not fully understood, and the involvement of androgen receptor signaling in prostatic inflammation may contribute to this complexity[48,49]. Reduced circulating androgen levels in aging men might contribute to increased inflammation in BPH, but whether the increase in inflammation is in response to a stimulus (e.g. wound repair) or is autoimmune-mediated remains to be determined. It is also possible that TNF secreted by inflammatory cells alters prostatic androgen/estrogen levels since this cytokine is known to stimulate aromatase activity; TNF-antagonist treatment in RA decreases aromatase action and increases synovial androgen levels[50].

Since BPH patients have one or more of a variety of histological features, including stromal nodules, glandular nodules, and fibrosis, this disease is more likely a combination of conditions[51]. Recent molecular profiling studies have provided a basis for at least two BPH subtypes[11]. As in prostate development, it has long been clear that paracrine interactions between stromal and epithelial cells are important in BPH[41]. These results of TNF-mediated stimulation of stromal, but not epithelial, growth in vitro support a hypothesis that stromal factors produced in response to TNF contribute to epithelial proliferation in vivo. However, the identities and function(s) of TNF-regulated, stromal-derived factors that modulate epithelial proliferation in BPH remain unknown. Of course, it is also possible that TNF supports epithelial proliferation in vivo through a direct effect not recapitulated in culture.

Although the local concentrations of TNF in the human prostate are not known, these studies used physiologically feasible concentrations of TNF in vitro based on reported secretions by inflammatory macrophages, even though circulating serum levels in humans are in the pg/mL range[52–55]. Adding to this complexity, appreciating a function for immune cells in this intercellular communication with stromal and epithelial cells opens new therapeutic targets and will be critical to elucidating mechanisms by which TNF or other signaling pathways drive hyperplasia and/or fibrosis. Reductions in numerous inflammatory or autoimmune-related pathways were observed by bulk RNA-seq after etanercept treatment. Upregulation of adhesion and muscle contraction pathways after TNF blockade could be interpreted as either greater stability within the microenvironment or induction of apoptosis through ROCK-mediated alterations in actin-myosin contractility,[56] but apoptosis was not directly interrogated in these studies. Whether inflammation drives a specific subtype of BPH is not clear, but it is certainly possible that inflammatory factors influence many features of BPH, including hyperplasia and collagen deposition[57–59].

The use of scRNA-seq analyses in both characterizing the cell types present and providing a molecular understanding of diseases have been extremely useful in moving biological studies forward. Evaluation of cell populations from synovial tissue in RA patients indicate that while different cell types may express either TNF or TNFRSF1A (TNFR1), only monocytes expressed high levels of both of these genes[31]. Furthermore, inflammatory monocytes in ulcerative colitis and Crohn's disease express TNF but may also aid in anti-TNF therapy resistance[32,33]. Even though

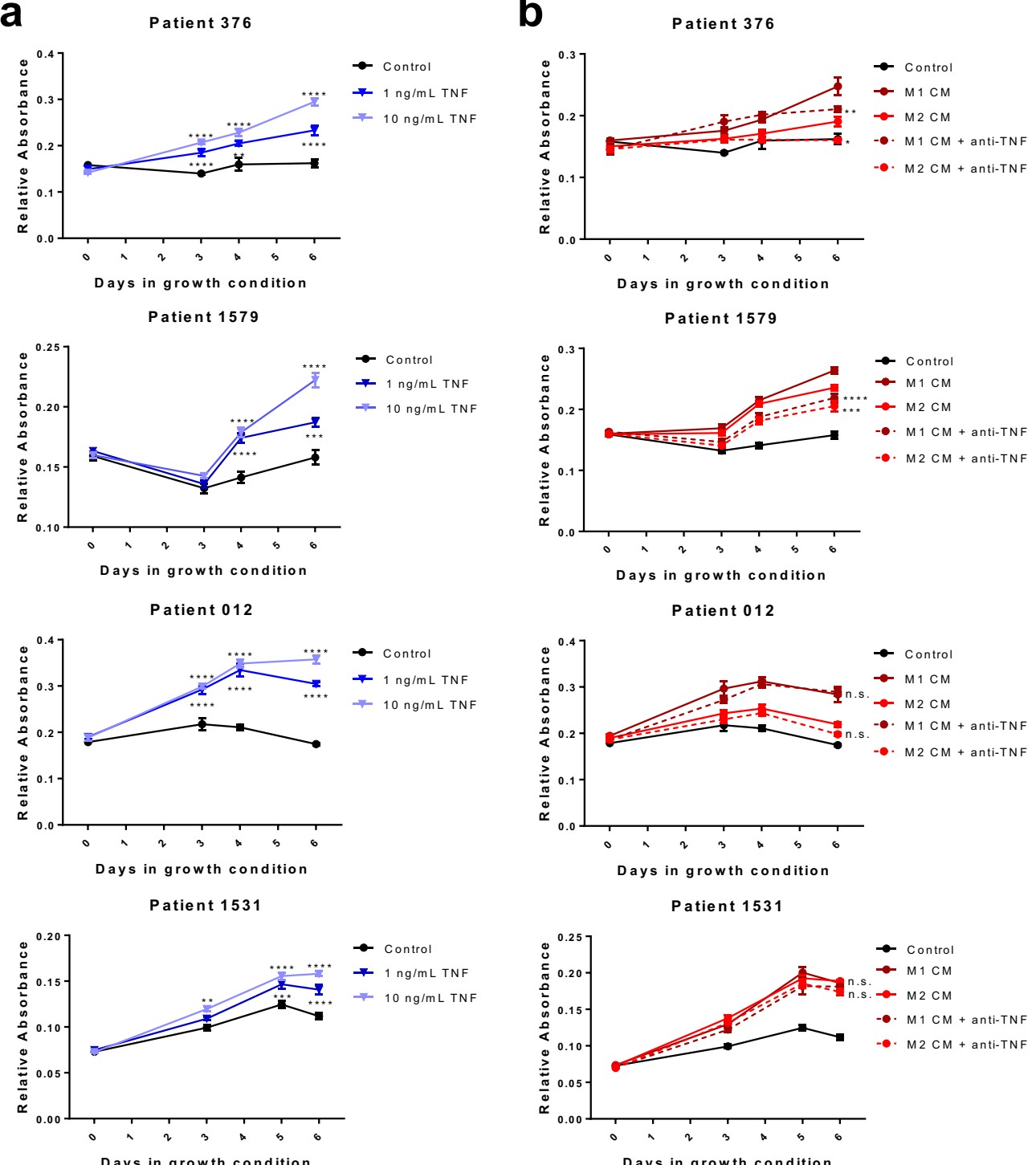

**Fig. 6 Macrophage-derived TNF promotes prostate fibroblast cell growth.** Primary prostate fibroblasts from two BPH tissues (patients 376, 1579, 012, and 1531) were subjected to indicated treatments. Crystal violet growth assays were performed in low serum conditions (0.5%) over six days. **a** Fibroblast cultures ($n = 4$ independent patients) were grown in the presence or absence of 1 or 10 ng/mL recombinant TNF (dark and light blue, respectively). Asterisks indicate significant differences compared to control samples (black lines), determined by two-sided, two-way ANOVA with multiple comparisons test. **b** Fibroblast cultures ($n = 4$ independent patients) were grown in the presence of 50% M1 (dark red) or M2 (light red) macrophage conditioned medium (generated from THP-1 cells) ±40 μg/mL TNF neutralizing antibody. Conditions containing anti-TNF neutralizing antibody are indicated with dashed lines. Points indicate the mean ± SEM of at least five technical replicates and graphs are representative of three independent experiments. Asterisks indicate significant differences compared to the paired conditioned medium condition without anti-TNF neutralization, using a two-sided, two-way ANOVA with Tukey's multiple comparisons test. Source data are provided as a Source Data file. n.s.=not significant, $*p < 0.05$, $**p < 0.01$, $***p < 0.001$, and $****p < 0.0001$.

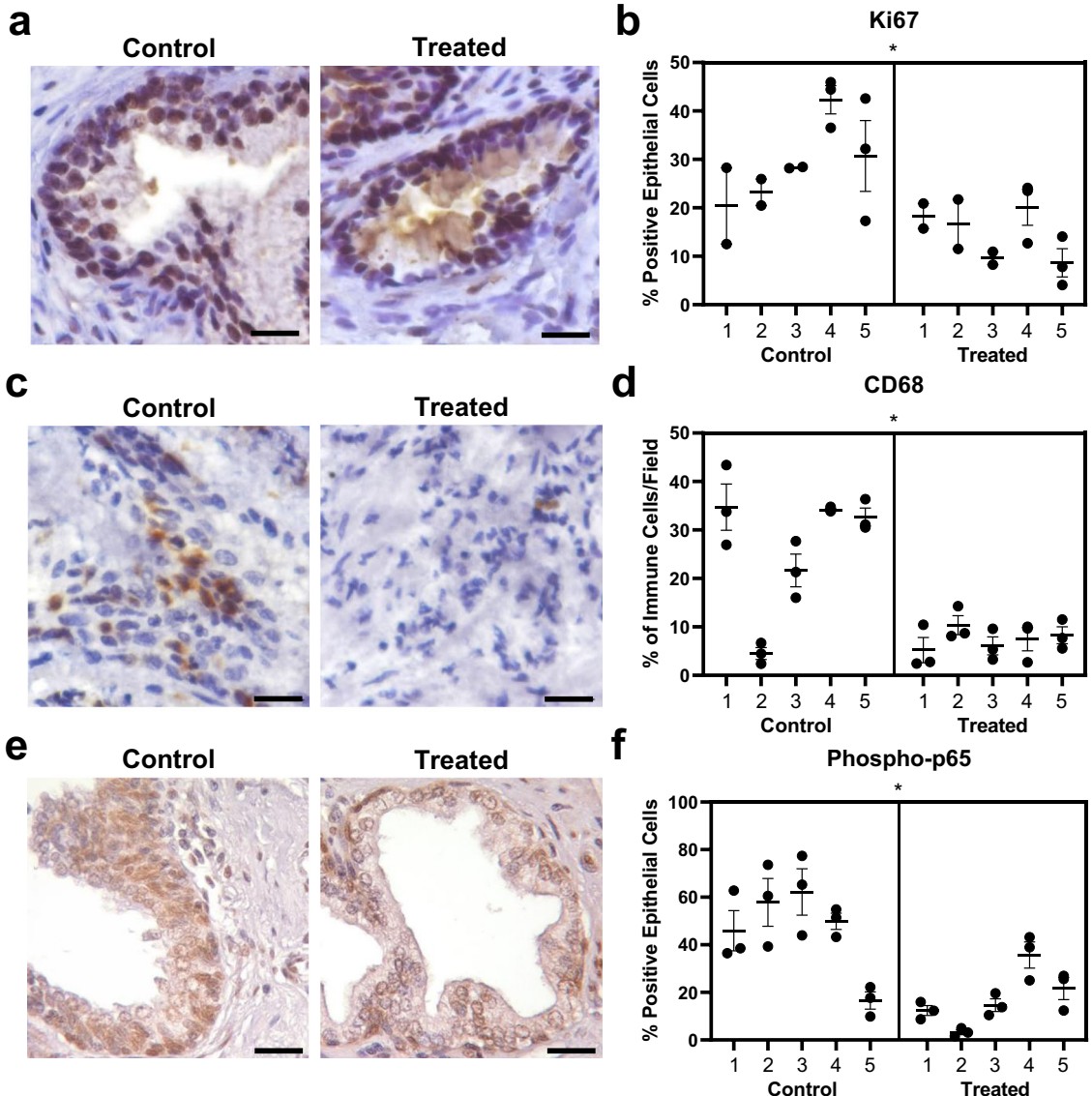

**Fig. 7 Patients treated with TNF-antagonists have decreased prostatic epithelial proliferation, macrophage infiltration, and NFκB activation.** Human transition zone tissues from patients taking TNF-antagonists at the time of radical prostatectomy or matched controls ($n = 5$ per group) were used for histological evaluation. **a** Representative images of IHC staining for Ki67 in control or treated patients, where brown color in the nucleus indicates positive staining. **b** Quantitation of IHC staining for epithelial Ki67 in control or treated patients, indicated as the percent Ki67$^+$ epithelial cells per field (*$p = 0.0123$). **c** Representative images of IHC staining for CD68 in control or treated patients, where brown color indicates positive staining. **d** IHC quantitation indicating the abundance of CD68$^+$ macrophages, represented as the portion of all immune cells per field (*$p = 0.0148$). **e** Representative images of phospho-p65 staining via IHC in control or treated patients, where brown color in the nucleus indicates positive staining. **f** Data represents the quantitation of IHC staining for epithelial phospho-p65 staining in control or treated patients, represented as the percent positive epithelial cells per field (*$p = 0.0172$). **b, d, f** Data indicate the mean ± SEM of the percent positive cells per field, where individual points indicate separate fields for each patient. Comparison of control and TNF-antagonist treated groups were conducted using a two-tailed nested $t$ test and asterisks indicate the significance of the treatment. Source data are provided as a Source Data file. Scale bars = 20 μm.

it is clear that TNF signaling is important in AI conditions and likely also in BPH, numerous cell types and/or inflammatory pathways can be explored for therapeutic potential using these scRNA-seq datasets. For example, data supports that TNF-activated macrophages are required for the pathogenesis of psoriasis, rather than other immune cell types, even though other cells contribute to the disease[60]. Our data also indicate a variety of immune cells exist in BPH and it is likely that each has a function. Given the significant reduction in antigen presentation pathways after etanercept treatment observed by bulk RNA-seq, it is likely that lymphocytes also contribute to BPH. Indeed, targeting antigen presentation is being pursued for treating various

autoimmune conditions[61,62]. Whether the inflammatory cell populations in BPH closely mirror those of an individual AI disease remains to be determined[31–33,63].

Due to the retrospective nature of these studies, there are several limitations. While this population represents a large cohort of men with BPH, the subset of patients undergoing treatment for AI disease with a concomitant diagnosis of BPH remains limited. The samples used were dependent on the surgical procedure and the prostate biorepository has tissues from less than 800 patients. Expanding the patient population will determine any capacity for TNF-antagonists to reduce surgical endpoints for BPH as well as discern if there are associations with

AI disease treatment and BPH diagnoses across racial and ethnic groups. Another limitation of these studies is that the specific cell types that contribute to BPH through direct versus indirect responses to TNF have yet to be determined in vivo. Obvious limitations using in vitro assays, including a lack of complexity compared to the in vivo BPH microenvironment, warrant further investigation of TNF function in the prostate. Additionally, no true murine model for BPH exists, so the NOD and Pb-PRL mouse models used for these studies lack some aspects of human disease (such as fibrosis and the influence of androgen:estrogen ratios) and may limit the translational potential of future mechanistic studies. Finally, since the occurrence of a prostatectomy in men taking TNF-antagonists is rare, scRNA-seq studies to date have not been able to directly evaluate the impact of these drugs on the prostate's inflammatory milieu and proliferative prostatic cell types.

It is possible these drugs would be most beneficial to patients with chronic prostatic inflammation, so it would also be useful to detect intraprostatic inflammation through non-invasive imaging procedures[64,65] or to be able to determine the risk of development of BPH through personalized medicine and genetic risk scores[10]. Further evaluation of human BPH tissues will determine if the presence of inflammatory macrophages or other identified cell signatures could contribute to anti-TNF therapy resistance[33]. The effect of TNF in stromal versus glandular BPH nodules also remains to be determined.

TNF-antagonists are already widely used and future work will determine their clinical utility in BPH patients. This includes whether short-term TNF-antagonist treatments (to limit the side effects of systemic anti-inflammatory agents) or if a combination with current BPH therapies may be beneficial. It is noteworthy that other immunomodulatory drugs with reduced long-term side effects might be considered for limiting BPH-related inflammation. Such studies will provide a basis for additional urological evaluation via earlier monitoring and therapeutic intervention in men with AI diseases. Tailoring the AI disease treatment strategy to include limiting BPH incidence could add therapeutic benefit for men. Such a strategy could both enhance patient quality of life and decrease the need for BPH surgical endpoints in these patients.

## Methods

All research studies were conducted in accordance with applicable local, state, and national regulations. Human studies were conducted under the approval of the NorthShore University HealthSystem Institutional Review Board (IRB). No patients received compensation for participation in these studies. Studies involving NOD mice were conducted with approval from the NorthShore University HealthSystem Institutional Animal Care and use Committee (IACUC), and studies involving Pb-PRL mice were conducted with approval from the Roswell Park Comprehensive Cancer Center IACUC.

**Enterprise database warehouse human subjects study**. A retrospective evaluation of patients was conducted for 112,152 males with office visits at NorthShore University HealthSystem between 01/01/2010 and 12/31/2012. The IRB-approved study provided deidentified data and did not require informed consent. A preliminary analysis determined that an expanded dataset should be used for final analysis. All ICD9/10 diagnosis codes, CPT procedure codes, and medication names used for data collection are included in Supplementary Tables 1–3. Patients under 40 years of age or with a diagnosis of prostate cancer were excluded from the study. Records were searched for patient diagnoses of BPH or any AI conditions (Supplementary Table 1). Baseline BPH incidence rates were compared with Chi-square tests. Chi-square tests were utilized to compare the proportion of men with BPH by autoimmune condition to the proportion of men with BPH and no AI condition, as well as to compare the proportion of BPH diagnoses in men being treated for an AI condition to the proportion of men with BPH and no AI condition. Predictors of BPH diagnosis were tested using multivariable logistic regression models adjusting for parameters of age, race, ethnicity, and BMI. Statistical significance was established throughout at $p < 0.05$. Statistical analyses were conducted using SAS version 9.4 (SAS Institute Inc, Cary, NC).

**Isolation of CD45$^+$ cells from human tissues**. Human prostate tissues were ethically procured with the IRB-approved NorthShore Urologic Disease Biorepository and Database with informed consent and deidentified clinical annotation. Small prostate tissues were obtained from ten male patients undergoing robotic-assisted laparoscopic prostatectomy (RALP) for prostate cancer, International Prostate Symptom Score (IPSS) of <15, Gleason 6-7, and estimated prostate volume of <40 grams by imaging with transrectal ultrasound (TRUS), CT scan, or MRI. Large prostate tissues were obtained from four male patients undergoing either RALP for prostate cancer (Gleason 6–7) or simple prostatectomy for BPH and had an estimated prostate size of >90 g. The patient ages ranged from 61 to 76 years. Tissues were pathologically verified to have no (or minimal) cancer burden. The transition zone (TZ) was dissected similarly for both small and large prostate tissues and separated for formalin-fixed paraffin-embedded (FFPE) histology or digested and prepared for fluorescence activated cell sorting (FACS). Tissues were minced, then digested while shaking at 37 °C for 2 h in 200 U/mL Collagenase I (Gibco) + 1 mg/mL DNase I (Roche) + 1% antibiotic/antimycotic solution in Hank's Balanced Salt Solution. Digestion solution was replaced with TrypLE Express dissociation reagent (Gibco) and allowed to shake at 37 °C for 5–10 min. Digested samples were neutralized in RPMI + 10% FBS, then mechanically disrupted by pipetting repeatedly. Samples were passed through a 100 μm cell strainer, then washed. Red blood cells were lysed in a hypotonic buffer, then cells were stained with Zombie Violet (Biolegend) and blocked with Human TruStain FcX blocking antibody (Biolegend). CD45-PE [clone HI30], EpCAM-APC [clone 9C4], and CD200-PE/Cy7 [clone OX-104] antibodies (Biolegend) were added to stain samples in preparation for FACS on a BD FACSAria II. Approximately 100,000 viable CD45$^+$CD200$^-$EpCAM$^-$ cells were sorted for downstream analysis[66].

A separate tube of the digested cell suspension was labeled for flow cytometry analysis of immune cells and stained with Zombie Violet (Biolegend) as well as CD45-FITC [clone HI30], CD11b-PE/Cy7 [clone ICRF44], CD19-APC/Cy7 [clone HIB19], CD3-APC [clone UCHT1], CD4-PE [clone RPA-T4], and CD8-BV510 [clone RPA-T8] antibodies. Information for all antibodies can be found in Supplementary Table 13.

**scRNA-seq of CD45$^+$ cells**. FACS-isolated cells were spun down and washed at least twice prior to loading onto the 10X Chromium System (10X Genomics), with Single Cell 3' Library & Gel Bead Kit, v3.0 reagents. Cells from three small and three large tissues were stained with TotalSeq-B Antibodies (Biolegend) for CITE-seq analysis. Antibodies for CD3 [clone UCHT1], CD4 [clone RPA-T4], CD8 [clone RPA-T8], CD19 [clone HIB19], and CD11b [clone ICRF44] were used following the manufacturer's instructions prior to loading into the Chromium System. Cells were loaded for downstream evaluation of 5000 cells/sample and cDNA amplification and library preparation were conducted according to the manufacturer's instructions. Libraries were sent to the Purdue Genomics Core Facility for post-library construction quality control, quantification, and sequencing. A high sensitivity DNA chip was run on an Agilent Bioanalyzer (Agilent) per the recommendation of 10x Genomics. Additional quality control was performed by running a denatured DNA pico chip (Agilent) followed by an AMPure cleanup (Beckman Coulter). Final library quantification was completed using a Kapa kit (Roche KK4824) prior to sequencing. Sequencing of normalized pools was conducted using a NovaSeq S4 flow cell on a NovaSeq 6000 system (Illumina) with 2x150 base-pair reads at a depth of 50,000 reads/cell. Libraries generated from cell surface protein labeling with TotalSeq-B antibodies were sequenced at a depth of 5000 reads/cell.

**scRNA-seq of all BPH cells**. Simple prostatectomy tissues from five BPH patients were minced and digested with 1000 U/mL Collagenase I + 1 mg/mL DNAse + 1% antibiotic/antimycotic for 4 h while shaking at 37 °C, followed by treatment with TrypLE Express reagent as above. After cell washing and RBC lysis, cell suspensions were subjected to debris removal with Debris Removal Solution (Miltenyi 130-109-398). Cells were washed and loaded into the Chromium System with v3.0 or v3.1 reagents for downstream evaluation of 10,000 cells/sample, followed by library preparation and quality control as above. Sequencing was performed on a NovaSeq S4 flow cell with 2x150 base-pair reads at a desired depth of 50,000 reads/cell.

**Data processing and quality control**. Sequencing reads from the Chromium system were de-multiplexed and processed using the CellRanger pipeline v3.0.0 (10x Genomics). CellRanger mkfastq was run to generate FASTQ files where dual indices were ignored, barcode mismatch allowance was set to 0, and the flag was set to indicate —use-bases-mask=Y26n*,I8n*,n*,Y98n. CellRanger count was then used for alignment, filtering, barcode counting, and unique molecular identifier (UMI) counting. All reads were aligned to the ENSEMBL human genome version GrCh38 using the STAR aligner v2.5.4[67]. CellRanger was run with the number of expected cells set to 5000.

R version 3.5.1 and Bioconductor version 3.8 were used in all statistical analyses. Cells that had fewer than 1,000 or greater than 10,000 observed genes were discarded. Cells were also removed if greater than 22% of all reads mapped to mitochondrial genes. Summaries of the data produced by the scRNA-seq analyses, including the run metrics, are shown in Supplementary Tables 14, 15.

**Unsupervised clustering and identification of marker genes**. Seurat version 3.1.3 was used for data normalization and cell clustering based on differential gene expression[68,69]. Data were normalized using scTransform[70] v.0.3.1 and cell cycle-related genes were used to produce a cell cycle score for each cell. Cell cycle scores, mitochondrial reads, and UMI counts were used to regress out heterogeneity from these variables by scaling the data. These corrected data, after permutation and selection of the first 30 principal components based on principal component analysis (PCA) scores, were used for downstream analysis. Unsupervised clustering was performed in Seurat, which uses graph-based approaches to first construct K-nearest neighbor graphs (K = 30) and identifies clusters by iteratively forming communities of cells to optimize the modularity function. The number of clusters were determined using the Louvain algorithm[71] for community detection, as implemented in Seurat with a resolution of 0.2. The correct resolution to use was determined both visually through plots and heat maps as well as using clustering trees via the clustree[72] R package v0.4.3, selecting a resolution that provides stable clusters. P-values were corrected for multiple testing using the Benjamini-Hochberg method[73]. Biomarkers were considered statistically significant at a 1% false discovery rate (FDR) using the Wilcoxon rank sum test[74]. Differentially expressed genes between small and large sample groups were identified using the edgeR[75,76] Bioconductor package, v3.31 with an FDR cutoff of 5%.

**Animal studies**. Male non-obese diabetic (NOD) mice on the inbred NOD/ShiLtJ background were purchased from The Jackson Laboratory (Bar Harbor, ME; Stock Number: 001976) and maintained in a barrier animal facility at NorthShore University HealthSystem. All studies were conducted according to US federal and state regulations and approved by the NorthShore IACUC (protocol #EH-15-064). Mice age 25 weeks were injected with 4 mg/kg TNF antagonist etanercept (Enbrel; $n = 12$) or vehicle (PBS, $n = 8$) twice weekly by intra-peritoneal injection for five weeks. Mice were randomly assigned to groups using block stratification. Diabetic status was tested using a Contour glucometer (Bayer 7151H)[21], at the time of first injection and last injection. Mice were harvested at 30 weeks.

Pb-PRL transgenic mice on the C56Bl/6J background were from Dr. Kindblom, Sahlgrenska University Hospital. All studies were performed in accordance with the National Institute of Health Guidelines for the Care and Use of Laboratory Animals and approved by the Roswell Park IACUC (#1308 M). Male Pb-PRL mice (20–22 months) were treated with 4 mg/kg TNF ligand trap, etanercept (Enbrel, $n = 5$), or with vehicle (PBS, $n = 6$) twice a week by intra-peritoneal injection for 12 weeks. The ventral prostate volume was measured every four weeks using high-resolution, high-frequency ultrasound (Vevo770 system, VisualSonics) with either the 710 or 704b scan head[40]. Mice were anesthetized in a chamber using 3% isoflurane and then placed in the transverse position on a heated imaging platform (Vevo Integrated Rail System III, VisualSonics) with continued anesthesia via nose cone. The abdomens of the mice were depilated and ultrasound gel (Aquasonic 100, Parker Laboratories) was applied. The location of the ventral prostate was identified and images were acquired using the VisualSonics software. Images were imported into Amira software (Visualization Sciences Group) for 3D volume reconstruction, where anatomic boundaries were set manually and volume was subsequently calculated with Amira. Data is presented as mean +/− SEM, normalized to pre-treatment volume ($151 \pm 10$ mm$^3$). One animal in the PBS group was removed from the analysis for week 12 due to rapid prostate swelling, determined on necropsy to likely have arisen from a hemorrhage. Thus, statistical analysis includes $n = 5$ for each group at week 12. In both animal models, urogenital tract tissues were harvested and prepared for FFPE histology.

**Immunohistochemistry**. FFPE tissue sections of 5 μm thickness were mounted on slides and prepared for immunohistochemistry (IHC). Sections were deparaffinized in xylene and rehydrated using gradient ethanol concentrations. Antigen retrieval was completed with Antigen Unmasking Solution (Vector H-3300) by microwaving for 20 min at 30% power. Staining was completed using the Universal VECTASTAIN standard or Elite ABC kit (Vector PK4000, PK6200), following the manufacturer's instructions. Detailed information on antibodies used for staining in human and mouse tissues is included in Supplementary Table 13. Quantitation of Ki67$^+$ or phospho-p65$^+$ cells was performed within the epithelial cell compartment and F4/80$^+$ (mouse) or CD68$^+$ (human) macrophages were evaluated within the immune cell compartment. All human and mouse IHC quantitation was blindly completed by S.E.C. by counting the indicated number of fields under the 40x objective.

**Autoantibody ELISA**. Serum from vehicle- or etanercept-treated NOD mice was collected at sacrifice and stored at −80 °C. Concentration of anti-etanercept antibodies were measured by ELISA with indicated modifications from[77]. Briefly, 5 μg/mL etanercept (0.1 M sodium carbonate buffer, pH=9.6) was coated on the surface of a 96-well plate overnight at 37 °C. Blocking was completed with 2% skim milk. Serial dilutions of anti-etanercept antibody (EMD Millipore, clone ETA63C8; 0-500 ng/mL) were used for the standard curve. Serial dilutions of serum samples (1:100-1:100,000) were included in triplicate for analysis. HRP-conjugated anti-mouse IgG secondary antibody (Cell Signaling) was used at 1:10,000 dilution, followed by the addition of Ultra TMB-ELISA substrate solution (Pierce) and

H$_2$SO$_4$ stop solution (ThermoFisher). Absorbance was detected on a SpectraMax Plus plate reader (Molecular Devices) at 450 nm with removal of background at 540 nm.

**Bulk RNA-seq**. Frozen prostate tissues from vehicle- ($n = 4$) or etanercept-treated ($n = 4$) NOD mice were thawed in TRIzol solution (Invitrogen), followed by tissue shredding with a Bio-Gen PRO200 Homogenizer (PRO Scientific Inc.) and RNA isolation per TRIzol's manufacturer recommendations. RNA cleanup was performed with the RNeasy Plus Mini Kit (Qiagen 74134) and shipped to Novogene (Sacramento, CA) for poly A selection and library preparation with the NEBNext Ultra II RNA Library Prep Kit for Illumina (New England BioLabs E7770), 2x150 sequencing on a NovaSeq 6000 PE150, followed by bioinformatics analysis. Read mapping was conducted with STAR v2.6.1d[67], differential gene expression analysis was performed with DESeq2 v1.26.0[78], and significantly differentially expressed genes were determined based on an adjusted $p$-value≤0.05. Kyoto Encyclopedia of Genes and Genomes (KEGG) was used for enrichment analysis and visualization of altered pathways.

**Cell culture**. THP-1 cells were purchased and authenticated from ATCC (STRB0424) and used within 20 passages of acquisition. BHPrE-1, NHPrE-1, and BHPrS-1 cell lines were isolated and cultured as benign epithelial and stromal prostatic cell models[79,80]. Authentication of BHPrE-1 (STRA3426), NHPrE-1 (STRA3441), and BHPrS-1 (STRB0418) cells was completed by ATCC and all experiments were conducted within 20 passages of testing.

THP-1 cells were cultured exactly as indicated by ATCC. THP-1 cells were differentiated for 24 h with 10 ng/mL PMA (Sigma), followed by either M1 polarization with 10 pg/mL LPS (Sigma) + 20 ng/mL IFNγ (Peprotech) for 24 h or M2 polarization with 20 ng/mL IL-4 (Peprotech) + 20 ng/mL IL-13 (Peprotech) for 72 h. After polarization was complete, serum free medium was added to M1 or M2 macrophages and incubated for 24 h. Conditioned medium was then harvested, filtered at 0.22 μm, and stored at −80 °C until it was used for growth assays.

Isolation of primary fibroblasts from deidentified BPH patient tissues 012, 376, 1579, and 1531. Fibroblasts were isolated using defined methods from freshly isolated human prostate transition zone tissue from simple prostatectomy[81]. Tissues were minced and enzymatically digested for 4 h with 250 U/mL Collagenase I (Worthington LS004196) + 112 U/mL Hyaluronidase (Sigma H3506) in complete RPMI medium. Digested cell suspensions were washed three times and fibroblasts were cultured for purification and used for all assays prior to passage 12.

For crystal violet growth assays, cells were allowed to adhere to 96-well plates prior to indicated treatments. Human recombinant TNF was purchased from Peprotech and TNF-neutralizing antibody (Fisher P300A) was pre-incubated at 40 μg/mL with 50% macrophage conditioned medium for 4–6 h. Cells were grown in low serum (0.1%) with treatment conditions for up to six days and cells fixed with 4% paraformaldehyde. Cells were stained with crystal violet and washed, then stain solubilized in 10% acetic acid to obtain relative absorbance values.

**Statistical analysis**. Univariate and bivariate analyses were run for all variables prior to our main patient database analyses. For categorical variables, Chi-square test and Fisher's exact test (for any frequency <5) were used. For continuous variables, $t$ tests (parametric) and Mann-Whitney tests (nonparametric) were used. Data distributions were examined and all variables were tested for linear or non-linear relationships; the necessary transformation and imputations were implemented based on the raw data distribution. Predictors of BPH incidence and diagnosis were conducted using both univariate and multivariable logistic regressions, adjusting for any likely covariates including age, race, ethnicity, and BMI. The effect size was estimated using odds ratio (OR) and 95% confidence interval (CI). Kaplan-Meier analysis was used to determine the probability that patients remained free of BPH diagnosis after the indicated treatments for diagnosed AI conditions. Statistical analyses were conducted using SAS version 9.4. Statistical significance of in vitro assays was completed using a two-way analysis of variance (ANOVA), patient characteristics were compared using a Student's $t$ test, and immune cell compartments were correlated using linear regression using Prism software version 7.05 (GraphPad). IHC counts were evaluated using a nested $t$ test (Prism, v8). A $p$-value of less than 0.05 was considered significant. In data figures, significance is indicated by $*p < 0.05$, $**p < 0.01$, $***p < 0.001$, and $****p < 0.0001$.

**Reporting summary**. Further information on research design is available in the Nature Research Reporting Summary linked to this article.

## Data availability

The scRNA-seq data is available in GEO under accession numbers GSE164695 (leukocytes) and GSE183676 (all cells). The bulk RNA-seq of NOD prostate tissues is available in GEO under accession number GSE183414. The raw EDW dataset is not available due to IRB-restrictions. Any further information about tissue resources and reagents associated with these studies should be directed to, and will be fulfilled by, the corresponding author upon reasonable request. Source data are provided with this paper.

## Code availability

The R scripts used to perform the scRNA-seq analysis are available at https://github.com/natallah/BPH_scRNAseq_NatureComm.git and Zenodo[82] through an Apache 2.0 license, allowing users to freely use the scripts for any purpose.

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

## Acknowledgements

The authors appreciate the assistance of Catherine Zhu in extracting medical record data, Philip Fitchev and Yana Filipovich for technical assistance, and the Purdue University Genomics Facility for their aid in scRNA-seq library normalization and sequencing. The authors are grateful to the NorthShore Biospecimen Repository and patients who have donated their tissue for research, without which much of this work would not have been possible. This work was funded by 1P20DK116185 (S.W.H. and T.L.R.) and R01DK117906 (S.W.H.) from NIDDK, the Purdue University Center for Cancer Research (NIH grant P30CA023168), the IU Simon Cancer Center (NIH grant P30CA082709), and the Roswell Park Comprehensive Cancer Center and imaging facility (NIH grants P30CA016056 and S10OD010393-01). This work was also generously supported by the Collaborative Core for Cancer Bioinformatics, the Walther Cancer Foundation, and the Rob Brooks Fund for Precision Prostate Cancer Care.

## Author contributions

S.E.C., O.E.F., and S.W.H. created the study concept; R.E.V., S.E.C., O.E.F., T.L.R., K.L.N., and S.W.H. designed research studies; R.E.V., J.P., P.T., C.Z., B.T.H., A.P.G., O.E.F., and S.W.H. were involved in acquisition of patient tissues or accessing medical records; B.L., C-H.W., J.S.P., and N.A.L. performed statistical and bioinformatics analyses; R.E.V. L.A-B., R.Z., B.L., V.G., M.G., O.E.F., G.M.C., M.M.B., T.S., and S.E.C. performed data acquisition and all authors contributed to data analysis and interpretation. R.E.V., S.E.C., and S.W.H. wrote the manuscript and all authors critically revised the manuscript.

## Competing interests

The authors declare no competing interests.
