## [Peer Review File · Nature Communications]

TNF is a Potential Therapeutic Target to Suppress Prostatic Inflammation and Hyperplasia in Autoimmune DiseaseEditorial Note: Parts of this Peer Review File have been redacted as indicated to maintain the confidentiality of unpublished data.

REVIEWER COMMENTS

Reviewer #1 (Remarks to the Author):

Vickman RE et al: TNF Alpha is a therapeutic target to suppress prostatic inflammation and hyperplasia in autoimmune disease

The authors present an interesting potential link between incidence of benign prostatic hyperplasia in patients with autoimmune (AI) disease and suggest the use of a TNF-alpha antagonist for the treatment of BPH. The authors evaluated EMR of 112, 152 of male patients above the age of 40. They stratified the data based on the presence of a variety of autoimmune diseases (n=10,769) and no autoimmune disease (n=101,383) and then assessed the incidence of BPH in both patient cohorts. Additionally, they attempted to correlate median days of hospital stay due to BPH-related prostate surgery to a protective effect of TNF-alpha antagonist treatment for AI disease. By single cell RNA sequencing approach (scrRNA-seq), the authors demonstrated that CD45+ T-cells and macrophages (5,000 cells) in patients with large prostates express higher levels of TNF-alpha and the two TNF receptors TNFRSF-1A and TNFRSF-1B. In a NOD mouse model, the authors demonstrated an increased NF-kB activation. Treatment with etanercept (Enbrel, a TNF-alpha antagonist) decreased Ki67 and macrophage infiltration in this model. In a probasin-prolactin mouse model of prostatic enlargement and inflammation, they observed Ki67 staining and prostate weight decrease but not NFkB activation or macrophage infiltration. In patients treated with a TNF-alpha antagonist (n=5) they observed decreased Ki67 staining and reduced CD68+ macrophages. Based on these and other studies, the authors conclude that patients with autoimmune disease have a heightened susceptibility to BPH and that TNF-alpha blockade could be repurposed to target inflammation and suppress BPH.

The authors have tested a provocative hypothesis regarding a potential link between autoimmune diseases and BPH but the study falls short of supporting their conclusion regarding the use of a TNF-alpha antagonist as a possible treatment for BPH.

1. The authors seem to make an assumption that all AI diseases will respond similar to a TNF-alpha antagonist treatment. It has been documented that treatment with TNF-alpha antagonists such as etanercept (the same TNF-alpha antagonist also used in this study) causes a 200% increased risk (HR=2) for developing ulcerative colitis (UC) and Crohn's disease (CD). This study by Korzenik J et al included 17,018 individuals with AI diseases who were exposed to TNF-alpha antagonists, and 63 308 individuals who were not. Reference: *Aliment Pharmacol Ther* 2019 Aug;50(3):289-294.

The present study by Vickman et al suggest that incidence of BPH is similarly increased in patients with Rheumatoid Arthritis (38%), ulcerative colitis (30.2%) and Crohn's disease (27.4%). Therefore, one wonders how in the context of treating one AI disease with TNF-alpha antagonist would decrease BPH, but then the same drug (etanercept), will increase the risk for UC or CD, and then based on the authors' data should also increase the risk for BPH. This is confusing.

2. Notwithstanding the conflicting clinical implications, the conclusion regarding the use of a TNF-alpha antagonist as a possible treatment for BPH, is based on only the data obtained from five to seven patients. It is unclear for which AI disease(s) were these patients being treated with a TNF-alpha antagonist and for how long? Did these patients have other comorbidities, or also receive treatment for BPH?

3. It is unclear how many total number of patients in the cohort of 10,769 patients (or even 1,495) patients actually received TNF-alpha antagonist treatment. For making comparisons, it is also important to take into consideration, did these patients receive multiple or a single type of treatment for their AI disease, and if so, then it would be a major confounding factor.

4. The authors have not clearly defined the clinical cohort and it is unclear if the patients' demographics were matched when making various comparisons. For example, the authors make a comparison between median hospital stay for patients who underwent surgery for progressive BPH disease. They have a relatively large group of men without TNF-alpha antagonist or methotrexate treatment, but the treated groups include only 7 (TNF-alpha antagonist) and 15 (methotrexate)

patients. Here again, it is unclear – were these patients treated before the diagnosis of the disease, or after (a seemingly important comparison based on the authors' data), for how long were they treated, patients' age, serum PSA, dose administered, prostate volume, Karnofsky score, comorbid conditions, other treatments taken in the past or concurrently with TNF-alpha antagonist or methotrexate, etc.

5. It is curious that in the large study cohort, the authors found only seven patients who were treated with a TNF-alpha antagonist and who also underwent surgery for a BPH diagnosis. Did all seven patients receive the same TNF-alpha antagonist? In their cohort of 10,769 cases (more precisely, 9,274 cases who developed AI before BPH) how many patients received a TNF-alpha antagonist treatment? In this cohort about 38% of the patients should have RA and at least some should have received a TNF-alpha antagonist – how did their clinical outcome compared to the non-AI group, patients with AI treated with other treatments?

It is unclear how the hospital stay compared between patients with AI disease receiving treatments other than TNF-alpha antagonist treatment, those who received TNF-alpha treatment or those receiving NSAIDs including low-dose aspirin? The data are also presented as median days, but with a small sample size, mean±Sd should be presented.

6. For scRNA-Seq it is unclear why the authors focused on immune cells, whereas, the title includes hyperplasia, which clinically means the growth of glandular prostate and fibrosis. The authors show that high dose TNF modestly induces prostate fibroblast growth. Interestingly, TNF-alpha antagonist did not affect the growth prostate epithelial cells. From the literature, TNF-alpha appears to have a dual role, the major being induction of cell death. How do the authors reconcile with differences in the effect of TNF-alpha on prostate epithelial and fibroblasts in cell culture studies and also the mouse models, where etanercept treatment decreased Ki67 staining in the prostate epithelium?

7. Discussion of the results should include if the levels of TNF-alpha in prostate tissues compare the levels at which TNF-alpha stimulates fibroblast growth (a modest effect) growth.

Methodology

1. It is unclear if all the 112,152 patients that were evaluated were they all initial diagnosis cases or did they receive treatment and for how long? If they were also being treated for BPH, then it will complicate the conclusion as to which treatment is influencing the outcome.

2. Patient characteristics should be better presented.

2. It is unclear which parameters were included in the multivariate logistic regression model.

3. The rationale for various statistical methods that were used for data analyses is unclear. In some places they used a chi-square test while in others they used the Mann-Whitney test. What sort of data distribution was observed? For a small size comparison a Fisher's exact test should be used.

In summary, although the authors have presented a relatively large amount of data, the study did not take into consideration the complexities of associated with AI diseases, notwithstanding the considerable side effects associated with TNF-alpha antagonists when suggesting TNF-alpha as a therapeutic target to suppress prostatic inflammation and hyperplasia in AI disease.

Reviewer #2 (Remarks to the Author):

Vickman et al present a study that leverages a large medical database of over 100,000 medical records alongside human derived tissue for single cell CITE-seq, mouse models and histochemistry to determine if aging men with autoimmune/inflammatory disease have increased precedence of benign prostatic hyperplasia (BPH). They further investigate in mouse models and retrospective metadata if the use of TNF-alpha antagonists decreases pathogenesis relevant to onset of BPH and

show that TNF-alpha antagonism significantly influenced prostate inflammation related to BPH pathogenesis. Ultimately this study is significant as the largest retrospective analysis with model systems to show the link between BPH pathogenesis and the TNF-alpha antagonism blockage and holds promise to motivate additional research toward slowing progression of BPH in high risk individuals with therapeutic repurposing.

The significance and novelty of the results presented is ample and justifies the overall interest to the greater community of researchers in the autoimmune/inflammatory, prostate and urology, and multi-systems inflammatory response immunology communities. The following are general comments about the study design that should be addressed in more detail for consideration in the manuscript:

1. The use of the retrospective data warehouse expedition was strong in revealing that TNF-a antagonists, excluding methotrexate decreased ORs for BPH and aided in study design and cohort composition downstream toward the claim that the pathogenesis of BPH was significantly slowed down by use of these therapies and motivated the CD45+ selection for enrichment of inflammatory cell types for the single cell CITE-seq data generation. Clearly there are many more inflammatory responsive cells associated with large prostate and high gleason score individuals than those with smaller and lower scores as shown in Figure 2.

That being said, the authors should note in more detail if the 5,000 CD45+ selected cells were from specific regions of the tissue (internal or external to the prostate) and if the bulk tissue was selected/biopsied in equivalent loci in the small versus enlarged prostate cases to assure no physical bias of inflammatory/immune infiltrate in the 4 large versus 10 small cases.

2. The authors should also describe how they accounted for the imbalance between 4 large and 10 small prostate cellular subsets in the data correction. Generally the single cell analyses methods used were appropriate and standard UMAP approaches with proper filtering and annotation, but it would be useful to see (as a supplementary figure) the annotations from patient to patient to provide a statistic on the diversity and heterogeneity of both inflammatory populations and/or immune populations from patient to patient especially in the N=4 large cases to assure there is no bias driven by individual outliers. (this is purely computation and would not require any additional data generation but would be valuable to the supplemental and readers as it addresses the fundamental question of inflammatory responsiveness diversity across patients in each subcompartment) This would go well with S2F.

3. Minor note that in Figure 2C, I would say 5000 cells / sample (even though its correctly stated in the caption) just so people appreciate in full that this is indeed 69,850 cells across 14 samples)

4. The fold expression levels for the TNF feature plots is significant and flows nicely in driving the focus on BPH macrophages (CD68+) and genes for expression and response to TNFa that motivate the Pb-PRL and NOD mouse models to follow. These results using Ki67 histochemistry are convincing, but would be stronger if at least bulk RNAseq data were generated from the tissues in the Pb-PRL and NOD mouse studies to enable a pathway/network model versus control/treated mice to determine genes and pathways associated with prostatic regression, TNFa antagonism, and/or fibroblast proliferation. Single cell approaches would be an interesting comparison but is likely not necessary as bulk RNASeq could be quite informative of these pathway perturbations. This data would enable network driven comparisons between mouse and human but also other AI diseases with bulk RNAseq data derived from similar mouse models in the community.

5. The authors did a fair assessment of study limitations in the discussion which is appreciated given access to human tissues prospectively. I think they have strong data showing inflammatory factors play a role in influencing BPH pathogenesis and would be interesting to include more discussion on these results in relation to other AI driven diseases if with similar findings outside of BPH as noted in the introduction, but in context of these findings.

Overall, this is a very clearly written manuscript that supports the experimental findings (both metadata and experimental molecular and mouse model data) with statistical analyses that are appropriate and results that do motivate the influence of AI disease and TNFa antagonism in

promoting or reducing BPH progression due to inflammation. My only major critique is listed above and mainly around lack of molecular data for the mouse models which would have been an additional interest for network models. The other items are all superficial and can be addressed in the text or Figures.

Reviewer #3 (Remarks to the Author):

In this manuscript, Renee E. Vickman et al. propose the potential use of TNF-blocking drugs to treat benign prostatic hyperplasia (BPH). Retrospective analysis of medical records led them to identify that BPH is highly prevalent in patients with autoimmune disease and that treatment with a TNF-blocking fusion protein such as etanercept reduced BPH incidence in those patients with a concomitant autoimmune disease. In human samples, the authors identify a reduced proliferation in epithelial cells, reduced numbers of macrophages, and lower levels of NF- κ B signaling in prostatic tissues for patients treated with etanercept. These results are modeled in vivo and in vitro. In vivo, two murine models of BPH are treated with etanercept and prostate volume, and immunohistochemistry to evaluate epithelial proliferation, macrophage infiltration, and NF- κ B signaling are reported. In vitro, primary prostate fibroblast and cell lines are treated with TNF or monocyte-conditioned media with or without a TNF-blocking monoclonal antibody, and the proliferation is determined by crystal violet.

Chronic inflammation and TNF have been previously associated with BPH, but this manuscript points to a critical and therapeutically actionable role of this cytokine. Therefore, the results reported in this manuscript may be clinically relevant. However, the manuscript contains several methodological flaws that should be addressed to support the conclusions.

1) Immunohistochemical analysis of fibroblast proliferation, macrophage infiltration, and NF- κ B signaling is only performed in a limited number of samples. In Figure 3, only 3 out of 5 mice are used. In figure 4, only 3 out of 12 in the treated group and 8 in the control group are displayed. In figure 6, only four patients in panels B and F are included, and only 3 in panel D. All samples included in the different experiments should be included.

2) Statistical analysis of the quantification of immunohistochemistry provides counterintuitive results. Could the authors double-check the results? The nested T-test available in the Graphpad Prism is a mixed model that could be applied to these data.

3) The number of fields analyzed for the quantification of immunohistochemistry is not consistent. In Figure 4 panel G, only a single field is analyzed in control 1. In Figure 4 panel E and Figure 6 panel B, two fields are used in more cases. The number should be increased in those panels, and the field selection should be standardized.

4) The authors claim that TNF-antagonist reduces prostate hyperplasia in the NOD mouse model. However, the data of the prostate volume is not shown in Figure 4. These data should be included.

5) The long-term use of etanercept in mouse is problematic due to the induction of mouse anti-drug antibodies. Could the author demonstrate that etanercept is functional at the end of the experiment in injected mice? Moreover, only the phosphorylation of p65 is used as a surrogate of TNF signaling. However, NF- κ B can be activated by many other cytokines. A stronger demonstration of TNF-blockade in the prostate of treated mice would be desirable.

6) The TNF-blocking monoclonal antibody only reduced the proliferation of prostate fibroblast derived from patient 376 but not from patient 012. Fibroblasts from more donors should be included to generalize this finding

Minor comments

1) Figure legends of Figure S2F and S2E should be corrected.

2) In Figure 3 panel 3, a data point is not introduced at week 12. The authors comment on the reason in the material and methods section. This explanation should also be introduced in the Results section. In any case, it would be desirable to introduce this data point.

RESPONSE TO REVIEWER COMMENTS

Reviewer #1 (Remarks to the Author):

Vickman RE et al: TNF Alpha is a therapeutic target to suppress prostatic inflammation and hyperplasia in autoimmune disease

The authors present an interesting potential link between incidence of benign prostatic hyperplasia in patients with autoimmune (AI) disease and suggest the use of a TNF-alpha antagonist for the treatment of BPH. The authors evaluated EMR of 112, 152 of male patients above the age of 40. They stratified the data based on the presence of a variety of autoimmune diseases (n=10,769) and no autoimmune disease (n=101,383) and then assessed the incidence of BPH in both patient cohorts. Additionally, they attempted to correlate median days of hospital stay due to BPH-related prostate surgery to a protective effect of TNF-alpha antagonist treatment for AI disease. By single cell RNA sequencing approach (scRNA-seq), the authors demonstrated that CD45+ T-cells and macrophages (5,000 cells) in patients with large prostates express higher levels of TNF-alpha and the two TNF receptors TNFRSF-1A and TNFRSF-1B. In a NOD mouse model, the authors demonstrated an increased NF-kB activation.

Treatment with etanercept (Enbrel, a TNF-alpha antagonist) decreased Ki67 and macrophage infiltration in this model. In a probasin-prolactin mouse model of prostatic enlargement and inflammation, they observed Ki67 staining and prostate weight decrease but not NFkB activation or macrophage infiltration. In patients treated with a TNF-alpha antagonist (n=5) they observed decreased Ki67 staining and reduced CD68+ macrophages. Based on these and other studies, the authors conclude that patients with autoimmune disease have a heightened susceptibility to BPH and that TNF-alpha blockade could be repurposed to target inflammation and suppress BPH. The authors have tested a provocative hypothesis regarding a potential link between autoimmune diseases and BPH but the study falls short of supporting their conclusion regarding the use of a TNF-alpha antagonist as a possible treatment for BPH.

1. The authors seem to make an assumption that all AI diseases will respond similar to a TNF-alpha antagonist treatment. It has been documented that treatment with TNF-alpha antagonists such as etanercept (the same TNF-alpha antagonist also used in this study) causes a 200% increased risk (HR=2) for developing ulcerative colitis (UC) and Crohn's disease (CD). This study by Korzenik J et al included 17,018 individuals with AI diseases who were exposed to TNF-alpha antagonists, and 63 308 individuals who were not. Reference: *Aliment Pharmacol Ther* 2019 Aug;50(3):289-294.

The present study by Vickman et al suggest that incidence of BPH is similarly increased in patients with Rheumatoid Arthritis (38%), ulcerative colitis (30.2%) and Crohn's disease (27.4%). Therefore, one wonders how in the context of treating one AI disease with TNF-alpha antagonist would decrease BPH, but then the same drug (etanercept), will increase the risk for UC or CD, and then based on the authors' data should also increase the risk for BPH. This is confusing.

The reviewer is correct in that it is not likely that patients with every AI disease respond the same way to TNF α -antagonists. In fact, we agree completely that the links between diseases are extremely complex and that therapies for one disease

could be at the detriment of another. In our initial submission, we pooled patients treated with different medications to minimize the complexity of data presentation, but the revised submission includes the breakdown of medication use in AI disease patients as well as the rate of BPH diagnosis in each of these diseases (Supplementary Tables 8 and 9 and text pages 6-7). Per this reviewer's specific example, our data does include patients diagnosed with more than one autoimmune condition, and some of the patients with Crohn's disease and ulcerative colitis were taking TNF α -antagonists. Whether some patients in our cohort were taking TNF α -antagonists and then later diagnosed with UC or CD was not assessed. While this question is interesting and more research is needed to understand the complexities of links between inflammation-associated diseases, it is beyond the scope of this manuscript.

2. Notwithstanding the conflicting clinical implications, the conclusion regarding the use of a TNF-alpha antagonist as a possible treatment for BPH, is based on only the data obtained from five to seven patients. It is unclear for which AI disease(s) were these patients being treated with a TNF-alpha antagonist and for how long? Did these patients have other comorbidities, or also receive treatment for BPH?

In this EDW study, treatment of BPH with TNF α -antagonists wasn't actually the goal. Our initial hypothesis was simply that patients diagnosed with AI diseases would also experience elevated levels of BPH. This led to the discovery that TNF treatment apparently provided a protective effect. Changes to the last paragraph of the introduction (text pages 3-4) have been incorporated to clarify that the most common therapeutics used for AI diseases were evaluated, and then, based on the data, TNF α -antagonists were utilized for downstream experiments.

The supplemental data describing the patients mentioned by the reviewer here suggests the possibility of decreased BPH progression toward surgery, but concrete conclusions cannot be made from our data due to the small sample size. Since our set of men with BPH who take TNF antagonists and progress to surgery is very small, it lacks statistical significance. This observation elicited some questions from the reviewers that cannot be addressed due to the lack of identifiable information for these individuals, therefore, we have decided to remove the supplemental table describing these data. Instead, we will include these data in future studies that contain larger patient numbers which may aid in determining whether TNF α -antagonists, or other inflammation-modifying therapeutic, are beneficial in limiting BPH progression toward surgery. However, we do find it intriguing, and worthy of further study, that so few men taking these drugs are present in our database of urology patients.

For the interest of this reviewer, we have included some details about the patients with BPH who progressed to surgery while taking TNF-antagonists and/or methotrexate. Of the 20 patients who had a diagnosis of BPH followed by surgery for BPH and were also taking either TNF-antagonists/methotrexate, all but two of these patients were diagnosed with an AI disease. These diseases were:

- patients with both TNF-antagonists + methotrexate = 1 with RA+Psoriasis+CD; 1 with RA+lupus
- TNF-antagonists only = 1 with CD+UC; 1 with CD+UC+OtherAI; 2 with psoriasis+CD; 1 with psoriasis
- methotrexate only = 1 with RA+UC+otherAI; 1 with CD+otherAI; 4 with RA only; 2 with psoriasis only; 1 with RA+psoriasis; 1 with RA+lupus; 1 with RA+other AI; 2 without AI diagnosis

It is clear that the patients who progressed to surgery for BPH did not consistently have any single AI disease, and the majority of these patients had more than one AI disease. All 20 of these patients who progressed to surgery were taking 5ARIs and/or alpha-blockers prior to surgery for BPH, which is the standard-of-care for these patients.

Other comorbidities were not included in the EDW study. All information that was pulled from electronic medical records for this analysis are listed in Supplementary Tables 1, 2, and 3 (text page 5). The authors believe this will aid in the reader's understanding of the EDW study.

3. It is unclear how many total number of patients in the cohort of 10,769 patients (or even 1,495) patients actually received TNF-alpha antagonist treatment. For making comparisons, it is also important to take into consideration, did these patients receive multiple or a single type of treatment for their AI disease, and if so, then it would be a major confounding factor.

The reviewer is correct that we were not clear enough in providing information about the breakdown of drug treatments in our data. Thus, we have added specific information about AI disease medications for all AI patients together, as well as separately for each AI disease in Supplementary Table 8 (text page 6). Some patients received more than one drug (i.e. TNF-antagonists + methotrexate), but we have only included patients who received one drug or the other for our statistical analysis in the corresponding table, Supplementary Table 9 (text page 7). The groups in Table 1 were simply identified as patients taking methotrexate or patients taking TNF α -antagonists, so these patient subgroups may have had additional treatment(s).

4. The authors have not clearly defined the clinical cohort and it is unclear if the patients' demographics were matched when making various comparisons. For example, the authors make a comparison between median hospital stay for patients who underwent surgery for progressive BPH disease. They have a relatively large group of men without TNF-alpha antagonist or methotrexate treatment, but the treated groups include only 7 (TNF-alpha antagonist) and 15 (methotrexate) patients. Here again, it is unclear – were these patients treated before the diagnosis of the disease, or after (a seemingly important comparison based on the authors' data), for how long were they treated, patients' age, serum PSA, dose administered, prostate volume, Karnofsky score, comorbid conditions, other treatments taken in the past or concurrently with TNF-alpha antagonist or methotrexate, etc.

We believe the reviewer is referring to the median days from BPH diagnosis to surgery, since median hospital stay was never assessed in our analysis. In these data, we did not match detailed patient demographics when making comparisons other than our indicated inclusion criteria. As addressed in the previous comment, Supplementary Tables 1-3 indicate what data was pulled from these patients, so we do not have additional data to present such as comorbidities, medications for unrelated conditions, Karnofsky score, etc.

5. It is curious that in the large study cohort, the authors found only seven patients who were treated with a TNF-alpha antagonist and who also underwent surgery for a BPH diagnosis. Did all seven patients receive the same TNF-alpha antagonist? In their cohort of 10,769 cases (more precisely, 9,274 cases who developed AI before BPH) how many patients received a TNF-alpha antagonist treatment? In this cohort about 38% of the patients should have RA and at least some should have received a TNF-alpha antagonist – how did their clinical outcome compared to the non-AI group, patients with AI treated with other treatments?

Yes, there were 7 patients treated with TNFa-antagonists after AI disease diagnosis who later progressed to surgery for BPH, although we have removed this information in our revision as detailed above.

We have included Supplementary Tables 8 and 9 (text pages 6-7) to show the breakdown of treatments for patients diagnosed with AI disease prior to BPH as well as to compare the rate of BPH diagnosis in patients treated with TNF-antagonists with either the baseline BPH prevalence of 20.3% or the rate of BPH diagnosis in AI patients not treated with medications. In both cases, the use of TNF-antagonists decreases the likelihood of BPH diagnosis in patients with AI, but breaking this down these numbers into individual AI disease categories reduced sample numbers significantly and, as a result, did not produce significant results.

It is unclear how the hospital stay compared between patients with AI disease receiving treatments other than TNF-alpha antagonist treatment, those who received TNF-alpha treatment or those receiving NSAIDs including low-dose aspirin? The data are also presented as median days, but with a small sample size, mean±Sd should be presented.

We did not pull data on NSAID medication use, thus cannot compare the use of NSAIDs with the use of TNFa-antagonists. There are some data available on NSAID use and BPH, as summarized in the discussion. The supplementary table demonstrating the median days between BPH diagnosis and surgery has been removed for the reasons described above, so the mean +/- SD is no longer relevant.

6. For scRNA-Seq it is unclear why the authors focused on immune cells, whereas, the title includes hyperplasia, which clinically means the growth of glandular prostate and fibrosis. The authors show that high dose TNF modestly induces prostate fibroblast growth. Interestingly, TNF-alpha antagonist did not affect the growth prostate epithelial cells. From the literature, TNF-alpha appears to have a dual role, the major being induction of cell death. How do the authors reconcile with differences in the effect of TNF-alpha on prostate epithelial and

fibroblasts in cell culture studies and also the mouse models, where etanercept treatment decreased Ki67 staining in the prostate epithelium?

We focused on inflammatory cells due to an interest in defining inflammatory cell types and understanding signaling pathways that are prominent and important in BPH tissues. We certainly acknowledge that evaluating all cell types will be essential to elucidating the complex interactions and responses across various cell subpopulations. While our efforts to characterize specific mechanisms among multiple cell types which drive BPH progression are ongoing, we have including our recently acquired scRNA-seq data from a small population (n=5) of BPH patients in which we evaluated all cell types. These data are presented in Supplementary Figure 5 (text page 9). In these results, macrophages still have significantly elevated expression of *TNFRSF1B*. Unfortunately, *TNFRSF1A* was not detected by sequencing in these samples, so we could not report the likely wider distribution of expression of this TNF receptor.

The reviewer does bring up an important and interesting point regarding cellular response to TNF-alpha. Previous studies highlight that a delicate balance in TNF signaling is important because switching between pro-survival NFkB activation and cell death is based on cellular context (Webster and Vucic review, *Front. Cell Dev.* 2020). Many of the authors on this manuscript also work in the prostate cancer field, where some cells are extremely sensitive to apoptosis by TNF. In our studies, low-dose TNF treatment did not induce cell death or slow proliferation in the indicated benign epithelial cell types and, interestingly, induced cell proliferation in stromal cells. Since our results also indicate that epithelial cell proliferation is reduced after TNF α -antagonist treatment, we reconcile our results with the hypothesis that stromal factors produced in response to TNF α act on epithelial cells to stimulate epithelial proliferation, particularly *in vivo*. Work to define these mechanisms is ongoing in our laboratory, but a description of this hypothesis was incorporated into the Discussion (text page 15).

7. Discussion of the results should include if the levels of TNF-alpha in prostate tissues compare the levels at which TNF-alpha stimulates fibroblast growth (a modest effect) growth.

Serum concentrations of TNF α in male patients are generally in the pg/mL range, while inflammatory macrophages, for example, secrete TNF α in a ng/mL range. Thus, we are using concentrations that are relevant to the local environment in the prostate. This information, with references, has been incorporated into the discussion section of the manuscript (text page 15).

Methodology

1. It is unclear if all the 112,152 patients that were evaluated were they all initial diagnosis cases or did they receive treatment and for how long? If they were also being treated for BPH, then it will complicate the conclusion as to which treatment is influencing the outcome.

The goals of this study required an evaluation of initial BPH diagnosis. There is a subset of patients who received treatment with either alpha-blockers or 5 α -reductase inhibitors (5ARIs) prior to BPH diagnosis, but BPH was not defined until it was clinically diagnosed because alpha-blockers or 5ARIs may be used for treatment of other conditions (for example, kidney stones or hair loss, respectively). These studies used clinical diagnosis of BPH as the direct indication for the disease, regardless of whether or not the patients had a diagnosis of one or more AI diseases.

2. Patient characteristics should be better presented.

We have incorporated additional supplemental information to better present patient characteristics, including patient age, race, and ethnicity (Supplementary Tables 4-6). We have also included more specific breakdowns of BPH diagnoses within specific AI diseases, in supplemental material as indicated above (all in text page 5).

2. It is unclear which parameters were included in the multivariate logistic regression model.

The parameters were age, race, ethnicity, and body mass index (BMI). This information has been incorporated into the text (pages 6 and 19).

3. The rationale for various statistical methods that were used for data analyses is unclear. In some places they used a chi-square test while in others they used the Mann-Whitney test. What sort of data distribution was observed? For a small size comparison a Fisher's exact test should be used.

We conducted various statistical tests based on the type and nature of the data. For categorical variables, chi-square tests and Fisher's exact tests (for any frequency cell <5) were used. For numerical variables, t-tests (parametric) and Mann-Whitney tests (nonparametric) were used. We have clarified this in the text (pages 26-27).

In summary, although the authors have presented a relatively large amount of data, the study did not take into consideration the complexities of associated with AI diseases, notwithstanding the considerable side effects associated with TNF-alpha antagonists when suggesting TNF-alpha as a therapeutic target to suppress prostatic inflammation and hyperplasia in AI disease.

Reviewer #2 (Remarks to the Author):

Vickman et al present a study that leverages a large medical database of over 100,000 medical records alongside human derived tissue for single cell CITE-seq, mouse models and histochemistry to determine if aging men with autoimmune/inflammatory disease have increased precedence of benign prostatic hyperplasia (BPH). They further investigate in mouse models and retrospective metadata if the use of TNF-alpha antagonists decreases pathogenesis relevant to onset of BPH and show that TNF-alpha antagonism significantly influenced prostate inflammation related to BPH pathogenesis. Ultimately this study is significant as the largest

retrospective analysis with model systems to show the link between BPH pathogenesis and the TNF-alpha antagonism blockage and holds promise to motivate additional research toward slowing progression of BPH in high risk individuals with therapeutic repurposing.

The significance and novelty of the results presented is ample and justifies the overall interest to the greater community of researchers in the autoimmune/inflammatory, prostate and urology, and multi-systems inflammatory response immunology communities. The following are general comments about the study design that should be addressed in more detail for consideration in the manuscript:

1. The use of the retrospective data warehouse expedition was strong in revealing that TNF-a antagonists, excluding methatrexate decreased ORs for BPH and aided in study design and cohort composition downstream toward the claim that the pathogenesis of BPH was significantly slowed down by use of these therapies and motivated the CD45+ selection for enrichment of inflammatory cell types for the single cell CITE-seq data generation. Clearly there are many more inflammatory responsive cells associated with large prostate and high gleason score individuals than those with smaller and lower scores as shown in Figure 2.

That being said, the authors should note in more detail if the 5,000 CD45+ selected cells were from specific regions of the tissue (internal or external to the prostate) and if the bulk tissue was selected/biopsied in equivalent loci in the small versus enlarged prostate cases to assure no physical bias of inflammatory/immune infiltrate in the 4 large versus 10 small cases.

The source of cells used for scRNA-seq analyses were prostatic transition zone tissues. The tissues were verified to be benign or have minimal cancer burden. Post-surgical tissues were isolated for downstream applications similarly for both small and large prostates, so no physical bias should have occurred. These studies did not use any tissue source external to the prostate. The authors have added minor changes to this section of the results and methods sections to help clarify this experimental setup (text pages 7 and 20). Also, this reviewer mentions the difference in inflammatory cells in large prostates with high Gleason versus small prostates with low Gleason – to this point, the authors want to reiterate that these tissues were all derived from patients with either low Gleason scores (6-7) in areas of the prostate outside of the sampled transition zone, or patients had no cancer with surgery specifically for benign disease (simple prostatectomy tissues). The large prostate patients who were selected had high IPSS versus the small prostate patients with low IPSS.

2. The authors should also describe how they accounted for the imbalance between 4 large and 10 small prostate cellular subsets in the data correction. Generally the single cell analyses methods used were appropriate and standard UMAP approaches with proper filtering and annotation, but it would be useful to see (as a supplementary figure) the annotations from patient to patient to provide a statistic on the diversity and heterogeneity of both inflammatory populations and/or immune populations from patient to patient especially in the N=4 large cases to assure there is no bias driven by individual outliers. (this is purely computation and would not require any additional data generation but would be valuable to the supplemental and readers as

it addresses the fundamental question of inflammatory responsiveness diversity across patients in each subcompartment) This would go well with S2F.

Our scRNA-seq analysis uses an anchor-based approach that was used to correct for batch effects. We also used a non-parametric Wilcoxon rank sum test which is known to be appropriate for handling unbalanced data. To illustrate there are no significant patient outliers that drive irrelevant conclusions, we have included a UMAP visualization in Supplementary Figure 2G demonstrating the significant overlap of individual samples (text page 8).

3. Minor note that in Figure 2C, I would say 5000 cells / sample (even though its correctly stated in the caption) just so people appreciate in full that this is indeed 69,850 cells across 14 samples)

Figure 2C has been adjusted to clarify there were 5,000 cells/sample.

4. The fold expression levels for the TNF feature plots is significant and flows nicely in driving the focus on BPH macrophages (CD68+) and genes for expression and response to TNF α that motivate the Pb-PRL and NOD mouse models to follow. These results using Ki67 histochemistry are convincing, but would be stronger if at least bulk RNAseq data were generated from the tissues in the Pb-PRL and NOD mouse studies to enable a pathway/network model versus control/treated mice to determine genes and pathways associated with prostatic regression, TNF α antagonism, and/or fibroblast proliferation. Single cell approaches would be an interesting comparison but is likely not necessary as bulk RNASeq could be quite informative of these pathway perturbations. This data would enable network driven comparisons between mouse and human but also other AI diseases with bulk RNAseq data derived from similar mouse models in the community.

The authors conducted a bulk RNA-seq experiment during the revision process to identify genes/pathways that may be associated with prostatic regression in NOD mice treated with TNF α -antagonists versus control treated mice. We had frozen NOD prostate material that we could use for this purpose, but unfortunately we did not have any frozen Pb-PRL prostate tissues for similar evaluation. We have incorporated the experimental methods and results into the manuscript text and produced a new Figure 5 to present these results (pages 11-12 and 25). To summarize, the bulk RNA-seq data from 4 etanercept-treated and 4 control samples were used to identify differentially expressed genes, then further evaluated with KEGG enrichment analysis. Pathways involved in antigen presentation or various inflammatory diseases were significantly downregulated in samples treated with etanercept. Etanercept-treated samples displayed upregulation in several pathways, notably focal adhesion and muscle contraction. Supplementary Figures 8-10 display a selection of these pathways. These data are presented in the main text and will allow for additional mechanistic studies in the future.

5. The authors did a fair assessment of study limitations in the discussion which is appreciated given access to human tissues prospectively. I think they have strong data showing inflammatory factors play a role in influencing BPH pathogenesis and would be interesting to include more

discussion on these results in relation to other AI driven diseases if with similar findings outside of BPH as noted in the introduction, but in context of these findings.

We have included some additional text applying the results of these experiments to other literature related to autoimmune diseases in the Discussion section, page 16. This extra discussion highlights a role for macrophages in orchestrating psoriasis as well as the use of targeting other cell types such as T/B cells, given that antigen presentation pathways were significantly downregulated in response to etanercept in the bulk RNA-seq studies (Figure 5).

Overall, this is a very clearly written manuscript that supports the experimental findings (both metadata and experimental molecular and mouse model data) with statistical analyses that are appropriate and results that do motivate the influence of AI disease and TNF α antagonism in promoting or reducing BPH progression due to inflammation. My only major critique is listed above and mainly around lack of molecular data for the mouse models which would have been an additional interest for network models. The other items are all superficial and can be addressed in the text or Figures.

Reviewer #3 (Remarks to the Author):

In this manuscript, Renee E. Vickman et al. propose the potential use of TNF-blocking drugs to treat benign prostatic hyperplasia (BPH). Retrospective analysis of medical records led them to identify that BPH is highly prevalent in patients with autoimmune disease and that treatment with a TNF-blocking fusion protein such as etanercept reduced BPH incidence in those patients with a concomitant autoimmune disease. In human samples, the authors identify a reduced proliferation in epithelial cells, reduced numbers of macrophages, and lower levels of NF- κ B signaling in prostatic tissues for patients treated with etanercept. These results are modeled in vivo and in vitro. In vivo, two murine models of BPH are treated with etanercept and prostate volume, and immunohistochemistry to evaluate epithelial proliferation, macrophage infiltration, and NF- κ B signaling are reported. In vitro, primary prostate fibroblast and cell lines are treated with TNF or monocyte-conditioned media with or without a TNF-blocking monoclonal antibody, and the proliferation is determined by crystal violet.

Chronic inflammation and TNF have been previously associated with BPH, but this manuscript points to a critical and therapeutically actionable role of this cytokine. Therefore, the results reported in this manuscript may be clinically relevant. However, the manuscript contains several methodological flaws that should be addressed to support the conclusions.

1) Immunohistochemical analysis of fibroblast proliferation, macrophage infiltration, and NF- κ B signaling is only performed in a limited number of samples. In Figure 3, only 3 out of 5 mice are used. In figure 4, only 3 out of 12 in the treated group and 8 in the control group are displayed. In figure 6, only four patients in panels B and F are included, and only 3 in panel D. All samples included in the different experiments should be included.

As requested, we have completed the staining for and display quantitative data from all mouse and patient samples from these studies and incorporated the changes into Figures 3, 4, and 7, as well as in Supplementary Figure 6.

2) Statistical analysis of the quantification of immunohistochemistry provides counterintuitive results. Could the authors double-check the results? The nested T-test available in the Graphpad Prism is a mixed model that could be applied to these data.

The authors have adjusted the statistical analysis for IHC quantitation to the nested t-test available in Graphpad Prism, v8. The results can be observed in Figures 3, 4, and 7, as well as in Supplementary Figure 6.

3) The number of fields analyzed for the quantification of immunohistochemistry is not consistent. In Figure 4 panel G, only a single field is analyzed in control 1. In Figure 4 panel E and Figure 6 panel B, two fields are used in more cases. The number should be increased in those panels, and the field selection should be standardized.

The number of fields has been made more consistent, such that three fields were quantified per sample in the vast majority of cases. The results can be observed in Figures 3, 4, and 7, as well as in Supplementary Figure 6.

4) The authors claim that TNF-antagonist reduces prostate hyperplasia in the NOD mouse model. However, the data of the prostate volume is not shown in Figure 4. These data should be included.

The reviewer is correct that the prostate volumes for NOD mice was not included. Unfortunately, upon harvesting animals from experiments for these studies, the weights of the urogenital tract were measured, but not the prostate specifically. However, we did have some prostate tissues that were dissected upon harvest and frozen, so we were able to bring out some prostate tissues to take the measurements. Only four tissues could be measured for each the control and etanercept groups and no significant difference in prostate volume was determined, but these data are presented in Supplementary Figure 7A (text page 10). We note in the manuscript that two selected mouse models are different. In the Pb-PRL model aged mice with enlarged prostates were used and these animals were treated for a longer period of time, allowing us to determine if drug treatment would reduce the size of the already enlarged prostates. The NOD model is less robust and has a number of comorbidities associated with its autoimmune status. Thus, our major question with this model was whether TNF α -antagonists would reduce cellular proliferation and inflammation in the prostate (which did occur).

5) The long-term use of etanercept in mouse is problematic due to the induction of mouse anti-drug antibodies. Could the author demonstrate that etanercept is functional at the end of the experiment in injected mice? Moreover, only the phosphorylation of p65 is used as a surrogate of TNF signaling. However, NF- κ B can be activated by many other cytokines. A stronger demonstration of TNF-blockade in the prostate of treated mice would be desirable.

The authors conducted an anti-etanercept ELISA on frozen serum samples from a subset of control or etanercept-treated NOD mice. No serum from PB-PRL mice was collected/stored for analysis. These data indicated that autoantibodies were being produced after 5 weeks of treatment, although numerous changes were still detected in the prostate, as reported. These results are presented in Supplementary Figure 7B (text pages 11, 24).

To address the latter point, we used the KEGG enrichment analysis of the bulk RNA-seq data. Visualization of the TNF Signaling Pathway supports that TNF blockade was successful in these tissues, especially downstream of TNFR2 (Supplementary Figure 8; text page 11-12).

6) The TNF-blocking monoclonal antibody only reduced the proliferation of prostate fibroblast derived from patient 376 but not from patient 012. Fibroblasts from more donors should be included to generalize this finding

To address this comment, we have doubled the number of patient-derived fibroblasts to illustrate the point in Figure 6. Thus, Figure 6 now has four primary cell lines each from an individual BPH patients who underwent a simple prostatectomy for BPH. These four samples were each treated with TNF α and with macrophage conditioned medium with or without TNF α neutralization to generalize the point that TNF α induces stromal cell growth and that TNF blockade abrogates the induction of macrophage-induced cell growth in a subset of patients. All of this information was incorporated into the Results section related to Figure 6 (text page 12).

Minor comments

1) Figure legends of Figure S2F and S2E should be corrected.

This figure legend has been corrected.

2) In Figure 3 panel 3, a data point is not introduced at week 12. The authors comment on the reason in the material and methods section. This explanation should also be introduced in the Results section. In any case, it would be desirable to introduce this data point.

This point has been introduced in Figure 3A.

REVIEWER COMMENTS

Reviewer #2 (Remarks to the Author):

The authors have amply addressed my original remarks successfully and I think the manuscript is significantly improved as a combination of the 3 Reviewer's comments.

Specifically, to my own comments, I appreciate the inclusion of the additional bulk RNAseq data related to the identification of specific genes associated with prostatic regression in the treated versus control treated mice. The use of available frozen material is sufficient and does motivate a better understanding of potential mechanism as shown in the new Figure 5 and supplemental dataset via the KEGG enrichment analysis that was added to the updated manuscript draft. The up- and down-regulated pathways make sense in treated samples including the upregulation of focal adhesion and muscle contractility.

Further, the additional inclusion of Supplemental Figure 2G for sample-to-sample overlap addresses the imbalance and variability questions toward patient-to-patient sample diversity.

Addition of text to flesh out single cell sample sourcing as well as the analysis methods used for the author's anchor-based approaches are all relevant and even points of clarity on sample batch analysis using the Gleason scoring and IPSS in large versus small prostate definitions and patient selection were useful. Further addition of Discussion text that clarifies the role of macrophages in response to etanercept using the bulk data in Figure 5 was also useful.

I think the clarity addressing Review 1's clinical and study cohort design questions was also useful in that many of the open-ended questions about negative results were still useful in better understanding how the authors came to current discussion points and conclusions. The only remaining concern is just the overall diversity of AI disease manifestation and responsiveness being broad and potential power through sampling limitations, but I still think the paper merits publication with the current modifications and am pleased with the responses to my questions/concerns. Therefore, I am in favor of publication.

Reviewer #3 (Remarks to the Author):

All my previous concerns have been successfully addressed.

Reviewer #4 (Remarks to the Author):

The revised manuscript is largely responsive to the previous review. However, some concerns remain which weaken the study and require a more complete response from the authors.

1. The concern was raised that, although the TNF-alpha antagonist (etanercept) may decrease BPH, it may also increase the risk for UC or CD, which might have the counter effect of actually increasing the risk for BPH. The authors respond with additional information (supplementary tables 8 and 9) regarding the breakdown of medication used in the study cohort but declined to perform a longitudinal analysis of the patients taking etanercept to determine whether, indeed, UC or CD incidence developed in these patients as 'beyond the scope of this manuscript'. This investigation would not be beyond the scope of this manuscript and is actually rather essential to assess whether treatment with etanercept is a viable choice for patients with AI-associated BPH. One

would assume that the Northshore clinical database could be easily interrogated for this information, and indeed that should be accomplished for this study.

2. The title of the manuscript is "TNF Alpha is a Therapeutic Target to Suppress Prostatic Inflammation and Hyperplasia in Autoimmune Disease" but present limited human patient data to support this assessment. Indeed, the authors state in their rebuttal that this wasn't the goal of the study. Therefore, it seems the title is a bit strident and should be tempered to 'may be a therapeutic target' since the study doesn't go far enough to 'prove' that it is a therapeutic target. Clearly, human clinical trials are needed for that.

3. Though additional demographic patient data is provided (supplementary tables 1-3) patient matching was not performed. Comorbidities were not considered. NSAID use was not assessed. As noted above, the data likely exists but wasn't pulled for this study. That response doesn't answer these concerns. In particular, the co-use of anti-inflammatories in conjunction with TNF α -antagonists (which is not reported in the literature) needs to be investigated given that the study is focused on the contribution(s) of autoimmune disease to BPH development and progression.

4. The response to the potential role of TNF α as differentially reducing epithelial cell proliferation but inducing fibroblast proliferation in cell cultures, primary cultures, and the two mouse models, is unclear. The authors are quite correct that TNF α likely plays different roles in different cell types and that its effects can be highly context dependent. The observation that treatment with TNF α has no effect on epithelial cellular proliferation in vitro doesn't help one understand why the Pb-PRL and NOD mice dosed with etanercept demonstrated significantly decreased prostate epithelial proliferation as measured by Ki67 staining. Attributing this phenomena to undefined stromal factors produced in response to TNF α that stimulate epithelial proliferation, and hence could be targeted by etanercept, is speculative at best. Perhaps a more mechanistic theory focused on expression patterns of the two TNF α receptors might be more insightful since they do signal differentially and are not expressed at similar levels in the various model systems used in these studies.

NCOMMS-21-10785A: RESPONSE TO REVIEWER COMMENTS

Reviewer #2 (Remarks to the Author):

The authors have amply addressed my original remarks successfully and I think the manuscript is significantly improved as a combination of the 3 Reviewer's comments.

Specifically, to my own comments, I appreciate the inclusion of the additional bulk RNAseq data related to the identification of specific genes associated with prostatic regression in the treated versus control treated mice. The use of available frozen material is sufficient and does motivate a better understanding of potential mechanism as shown in the new Figure 5 and supplemental dataset via the KEGG enrichment analysis that was added to the updated manuscript draft. The up- and down-regulated pathways make sense in treated samples including the upregulation of focal adhesion and muscle contractility.

Further, the additional inclusion of Supplemental Figure 2G for sample-to-sample overlap addresses the imbalance and variability questions toward patient-to-patient sample diversity.

Addition of text to flesh out single cell sample sourcing as well as the analysis methods used for the author's anchor-based approaches are all relevant and even points of clarity on sample batch analysis using the Gleason scoring and IPSS in large versus small prostate definitions and patient selection were useful. Further addition of Discussion text that clarifies the role of macrophages in response to etanercept using the bulk data in Figure 5 was also useful.

I think the clarity addressing Review 1's clinical and study cohort design questions was also useful in that many of the open-ended questions about negative results were still useful in better understanding how the authors came to current discussion points and conclusions. The only remaining concern is just the overall diversity of AI disease manifestation and responsiveness being broad and potential power through sampling limitations, but I still think the paper merits publication with the current modifications and am pleased with the responses to my questions/concerns. Therefore, I am in favor of publication.

The authors appreciate that our efforts to improve this manuscript were received positively by this reviewer. Regarding this reviewer's final concern, it should be stated that the authors agree completely that AI diseases in general are quite diverse and are, indeed, a combination of unique diseases. For our analyses, the need to combine these diseases to reach sample sizes large enough for the purposes of experimental power could be a concern. Based on a comment from Reviewer #4 below, we split out a group of patients for additional analyses which we believe addresses this concern from Reviewer #1. We have added additional analyses in the form of Kaplan-Meier survival curves to demonstrate the probability of remaining free of a BPH diagnosis over time. This analysis was performed with all AI conditions combined, without Crohn's and ulcerative colitis (both forms of inflammatory bowel disease), or only including Crohn's or UC patients. These analyses demonstrate that TNF α -antagonists continue to significantly reduce the likelihood of BPH diagnosis in all three groups of patients (UC+CD patients; AI patients other than UC and CD; any AI condition). These data have been integrated into the manuscript and are shown in supplementary figures 1-3.

Reviewer #3 (Remarks to the Author):

All my previous concerns have been successfully addressed.

The authors greatly appreciate the reviewer's suggestions, as they significantly improved the quality of this manuscript.

Reviewer #4 (Remarks to the Author):

The revised manuscript is largely responsive to the previous review. However, some concerns remain which weaken the study and require a more complete response from the authors.

1. The concern was raised that, although the TNF-alpha antagonist (etanercept) may decrease BPH, it may also increase the risk for UC or CD, which might have the counter effect of actually increasing the risk for BPH. The authors respond with additional information (supplementary tables 8 and 9) regarding the breakdown of medication used in the study cohort but declined to perform a longitudinal analysis of the patients taking etanercept to determine whether, indeed, UC or CD incidence developed in these patients as 'beyond the scope of this manuscript'. This investigation would not be beyond the scope of this manuscript and is actually rather essential to assess whether treatment with etanercept is a viable choice for patients with AI-associated BPH. One would assume that the Northshore clinical database could be easily interrogated for this information, and indeed that should be accomplished for this study.

Given the reiteration of the importance of this concern, the authors have looked more closely at the impact of TNF α -antagonists on the diagnosis of specific AI conditions as well as the impact of medications on risk of BPH diagnosis in these subcategories of patients. The authors acknowledge this is an important consideration in the context of the presented work. Per the reviewer's request, we have conducted a longitudinal analysis to examine the association of medication use with BPH risk in three cohorts: patients with UC or CD, patients with AI disease other than UC or CD, and all AI disease patients. In this analysis, patients treated with TNF α -antagonists illustrated the highest BPH free survival in all three cohorts: patients with UC or CD (Supplementary Figure 1), AI disease patients without UC or CD (Supplementary Figure 2), and all AI disease patients (Supplementary Figure 3). We also examined of risk of increased UC or CD by various types of medication use. As shown in Supplementary Tables 11-12, patients treated with TNF α -antagonists had higher risk of UC, but not CD, compared to use of other medications. As expected, patients not treated with any medication had the highest incidence of UC and CD (Supplementary Tables 11-12).

2. The title of the manuscript is "TNF Alpha is a Therapeutic Target to Suppress Prostatic Inflammation and Hyperplasia in Autoimmune Disease" but present limited human patient data to support this

assessment. Indeed, the authors state in their rebuttal that this wasn't the goal of the study. Therefore, it seems the title is a bit strident and should be tempered to 'may be a therapeutic target' since the study doesn't go far enough to 'prove' that it is a therapeutic target. Clearly, human clinical trials are needed for that.

The authors agree that they have not proven that TNF α is a therapeutic target in clinical BPH since clinical trials were not included in these studies. Thus, it may not be appropriate to use such strong language and the authors have adjusted the title of the manuscript to read “TNF Alpha may be a Therapeutic Target to Suppress Prostatic Inflammation and Hyperplasia in Autoimmune Disease.” The authors have “toned down” the language related to this point throughout the manuscript.

3. Though additional demographic patient data is provided (supplementary tables 1-3) patient matching was not performed. Comorbidities were not considered. NSAID use was not assessed. As noted above, the data likely exists but wasn't pulled for this study. That response doesn't answer these concerns. In particular, the co-use of anti-inflammatories in conjunction with TNF α -antagonists (which is not reported in the literature) needs to be investigated given that the study is focused on the contribution(s) of autoimmune disease to BPH development and progression.

The authors realize that there may be comorbidities or other clinical factors that may be involved in BPH risk but were not assessed in these studies. This often occurs in retrospective analyses, especially when investigating many disease entities. Even though patient matching was not performed because of the striking variety of combinations of AI diseases within the patient population studied, we have included another supplementary table to provide additional information on how many patients had each combination of AI diseases (Supplementary Table 7). This table highlights both the diversity of AI diseases as well as the variety of co-occurrences of AI diseases in patients.

It is true that NSAID use is of interest to study in conjunction with TNF α -antagonists. Unfortunately, the co-use of these drugs is not possible to study within our experimental design or within our currently available data. While we have access to prescription drug information, the majority of NSAIDs used by urology patients are over-the-counter medications purchased and used under medical advice. As such, these drugs are rarely and unreliably included in medical record data. Knowing this, even though these drugs are relevant to these studies, we did not include NSAID use in our EDW study in Figure 1 and, as a result, the authors do not have this information in their completely de-identified database. The use of NSAIDs alone as an approach to counteract lower urinary tract symptoms is associated with considerable published literature, to which the authors refer in the Discussion section (reference numbers 42-43 in manuscript). These studies appear to show short-term, rather than long-term, effects and no profound alterations of outcome. Clearly, more research is needed in this area, and the authors have included a statement related to this point in the first paragraph of the discussion (page 15).

4. The response to the potential role of TNF α as differentially reducing epithelial cell proliferation but inducing fibroblast proliferation in cell cultures, primary cultures, and the two mouse models, is unclear.

The authors are quite correct that TNF α likely plays different roles in different cell types and that its effects can be highly context dependent. The observation that treatment with TNF α has no effect on epithelial cellular proliferation *in vitro* doesn't help one understand why the Pb-PRL and NOD mice dosed with etanercept demonstrated significantly decreased prostate epithelial proliferation as measured by Ki67 staining. Attributing this phenomena to undefined stromal factors produced in response to TNF α that stimulate epithelial proliferation, and hence could be targeted by etanercept, is speculative at best. Perhaps a more mechanistic theory focused on expression patterns of the two TNF α receptors might be more insightful since they do signal differentially and are not expressed at similar levels in the various model systems used in these studies.

This is a very intriguing question, which we are keen to address as a primary research direction coming out of this study. We have ongoing studies in this area, but these experiments and results have reinforced that the mechanisms regulating this signaling pathway is complex. Cell type specificity in a multicellular microenvironment can be difficult to recapitulate with *in vitro* methodology. The authors' discussion of the possibility of stromal factors that stimulate epithelial proliferation in response to TNF α truly is a speculation. The authors are not yet clear on why there is clear stimulation of stromal cells by TNF α *in vitro*, but histological evaluation of treatment with TNF α -antagonists *in vivo* seems to have more of an effect on epithelial proliferation. We must obviously remain aware that cells in 2D culture do not necessarily reflect what is happening *in vivo*. Nonetheless, the authors are conducting both co-culture and conditioned medium assays to attempt to recapitulate the possible crosstalk between fibroblasts and epithelial cells *in vivo*. So far, results have not been conclusive and are not ready for publication. To include the possibility that TNF α could result in a more direct stimulation of epithelial proliferation *in vivo*, we have included an additional sentence in the 4th paragraph of the discussion (page 16).

[REDACTED]

[REDACTED]

This experiment is quite complex and on its own does not directly identify a specific mechanism. The authors wish to use this as preliminary data in support of funding proposals rather than introduce a complicated experiment with no mechanistic conclusions into this manuscript. This mechanism will be interrogated in ongoing and future studies.

The reviewer's recommendation of evaluating the two TNF α receptors is a valid one, but still may not reflect the downstream signaling activities in various cell types. The authors did evaluate TNFR1 and TNFR2 expression in the control and TNF α -antagonist treated patient samples from Figure 7. Unfortunately, the heterogeneity both within and between samples made it very difficult to reach a conclusion as to whether TNF α -antagonist treatment altered TNF receptor expression. As a result, the authors do not believe the results merit inclusion in this manuscript.

[REDACTED]

[REDACTED]

[REDACTED]

The authors did, however, include additional data that suggests epithelial cells do not respond directly to macrophage conditioned medium in 2D culture, unlike the stimulation of fibroblast growth by macrophage conditioned medium. These new results are included in Supplementary Figure 15.

REVIEWERS' COMMENTS

Reviewer #4 (Remarks to the Author):

1. The response to concern 1 is stellar and the reviewer thanks the authors for providing the longitudinal analysis of the association between medication use with BPH risk.
2. The modification of the manuscript title is appropriate.
3. The authors provide a thoughtful and well-taken response to the query regarding NSAID use. Accurate measurement of NSAID use outside of actual clinical trials is inaccurate and untrustworthy, as stated. The reviewer thanks the authors for including Supplementary Table 7 to address the query considering the spectrum and combination of autoimmune disease in the patient cohort(s).
4. The data shown in new Supplementary Figure 15 partially responds to the query concerning the identities and role(s) of TNFa-regulated stromal-derived factors that stimulate (or inhibit) epithelial proliferation in the prostate. The revised text on p16, paragraph 4, in the Discussion section could more directly address the larger question by direct by stating that the identities and role(s) of TNFa-regulated stromal-derived factors that stimulate (or inhibit) epithelial proliferation in the prostate are not known and remain to be identified. This would reduce the 'speculative' nature of the statement and point towards the need for further studies (which the authors are undertaking) to better understand this phenomena.

NCOMMS-21-10785B

Response to Reviewers Comments

Reviewer #4 (Remarks to the Author):

1. The response to concern 1 is stellar and the reviewer thanks the authors for providing the longitudinal analysis of the association between medication use with BPH risk.

The authors agree that the incorporated data strengthened the manuscript overall.

2. The modification of the manuscript title is appropriate.

The authors appreciated the suggestion.

3. The authors provide a thoughtful and well-taken response to the query regarding NSAID use. Accurate measurement of NSAID use outside of actual clinical trials is inaccurate and untrustworthy, as stated. The reviewer thanks the authors for including Supplementary Table 7 to address the query considering the spectrum and combination of autoimmune disease in the patient cohort(s).

The authors agree and thank the reviewer for understanding the limitations of incorporating NSAID use.

4. The data shown in new Supplementary Figure 15 partially responds to the query concerning the identities and role(s) of TNFa-regulated stromal-derived factors that stimulate (or inhibit) epithelial proliferation in the prostate. The revised text on p16, paragraph 4, in the Discussion section could more directly address the larger question by direct by stating that the identities and role(s) of TNFa-regulated stromal-derived factors that stimulate (or inhibit) epithelial proliferation in the prostate are not known and remain to be identified. This would reduce the 'speculative' nature of the statement and point towards the need for further studies (which the authors are undertaking) to better understand this phenomena.

The authors agree that the previous discussion on this subject remained speculative, so a sentence has been incorporated in paragraph 4 of the Discussion section as suggested. This sentence helps directly indicate work that remains to be done.